# A marine sponge-derived lectin reveals hidden pathway for thrombopoietin receptor activation

Hiromi Watari [1,10], Hiromu Kageyama[2,10], Nami Masubuchi[3], Hiroya Nakajima[1], Kako Onodera[2], Pamela J. Focia [4], Takumi Oshiro[5], Takashi Matsui [5,6], Yoshio Kodera[5,6], Tomohisa Ogawa [2], Takeshi Yokoyama[2], Makoto Hirayama[7], Kanji Hori[7], Douglas M. Freymann [4], Misa Imai[3], Norio Komatsu[3,8,9], Marito Araki [3,8] ✉, Yoshikazu Tanaka[2] ✉ & Ryuichi Sakai [1] ✉

N-glycan-mediated activation of the thrombopoietin receptor (MPL) under pathological conditions has been implicated in myeloproliferative neoplasms induced by mutant calreticulin, which forms an endogenous receptor-agonist complex that traffics to the cell surface and constitutively activates the receptor. However, the molecular basis for this mechanism is elusive because oncogenic activation occurs only in the cell-intrinsic complex and is thus cannot be replicated with external agonists. Here, we describe the structure and function of a marine sponge-derived MPL agonist, thrombocorticin (ThC), a homodimerized lectin with calcium-dependent fucose-binding properties. In-depth characterization of lectin-induced activation showed that, similar to oncogenic activation, sugar chain-mediated activation persists due to limited receptor internalization. The strong synergy between ThC and thrombopoietin suggests that ThC catalyzes the formation of receptor dimers on the cell surface. Overall, the existence of sugar-mediated MPL activation, in which the mode of activation is different from the original ligand, suggests that receptor activation is unpredictably diverse in living organisms.

The thrombopoietin (TPO) receptor MPL plays critical roles in hematopoietic stem cell (HSC) maintenance and platelet production[1–3]. MPL, which lacks kinase activity, is activated via Janus kinase 2 (JAK2) bound to the intracellular domain of MPL. The detailed process of receptor dimerization and activation of JAK2 by TPO is poorly defined, partially due to the lack of structural information on MPL[4]. Under pathological conditions, myeloproliferative neoplasms (MPNs), a mutant form of the glycan-dependent molecular chaperone CALR (CALRmut), bind to the immature sugar chain of MPL in the endoplasmic reticulum and then translocate to the cell membrane to form a complex with functional MPL[5–7]. This complex leads to the transformation of hematopoietic cells in an MPL-dependent manner[8–10]. CALRmut activates MPL only when CALRmut binds to the immature sugar chain of the receptor and traffics to the cell surface[6,7,10,11]. However, the external agonist required for the recapitulation of this

[1]Graduate School of Fisheries Sciences, Hokkaido University, Hakodate, Japan. [2]Graduate School of Life Sciences, Tohoku University, Sendai, Japan. [3]Laboratory for the Development of Therapies against MPN, Juntendo University Graduate School of Medicine, Tokyo, Japan. [4]Department of Biochemistry & Molecular Genetics, Feinberg School of Medicine, Northwestern University, Chicago, USA. [5]Department of Physics, School of Science, Kitasato University, Sagamihara, Japan. [6]Center for Disease Proteomics, School of Science, Kitasato University, Sagamihara, Japan. [7]Graduate School of Integrated Sciences for Life, Hiroshima University, Higashi-Hiroshima, Japan. [8]Department of Advanced Hematology, Juntendo University Graduate School of Medicine, Tokyo, Japan. [9]Department of Hematology, Juntendo University Graduate School of Medicine, Tokyo, Japan. [10]These authors contributed equally: Hiromi Watari, Hiromu Kageyama. ✉e-mail: m-araki@juntendo.ac.jp; yoshikazu.tanaka@tohoku.ac.jp; ryu.sakai@fish.hokudai.ac.jp

mode of activation is not known. Therefore, the molecular basis of receptor activation by lectin-type ligands remains largely unstudied.

We recently identified a marine sponge-derived 14-kDa protein, thrombocorticin (ThC), as a potent agonist of MPL[12]. Here, we report the three-dimensional structure of ThC as a fucose-binding lectin and the mechanisms underlying its MPL activation by binding to sugar chains on MPL.

## Results

### Biochemical profiles of ThC

We identified a complete 131-amino acid sequence of native ThC (nThC) isolated from the sponge (Fig. 1A (i)) by mass spectrometry, Edman degradation after peptic digestion, LC−MS/MS and structural analysis through X-ray crystallography (Supplementary Note 1 and Supplementary Figs. 1–7). According to the determined sequence, N-terminal His-tagged recombinant ThC (rThC) promoted the proliferation of Ba/F3-HuMpl cells (Fig. 1B) in an MPL-dependent manner (Supplementary Fig. 8). Immunoblot analyses indicated that rThC activated steady-state JAK/signal transducer and activator of transcription (STAT) signaling (Fig. 1C). These data indicated that rThC activated MPL to promote cell proliferation in Ba/F3-HuMpl cells.

### Critical role of sugar-binding capacity in ThC-dependent MPL activation

The amino acid sequence of ThC shares approximately 33% identity with bacterial fucose-binding lectins. However, little similarity was found between ThC and TPO in amino acid sequences (Fig. 1A (ii)) and three-dimensional structures. The activation of MPL by lectin-like molecules that are structurally distinct from TPO attracted our

attention because the CALRmut interaction with an immature sugar chain attached to the receptor during receptor maturation was proposed as an alternative and pathological mode of MPL activation in MPN[5–7]. Therefore, we hypothesized that ThC binds to sugar chains on the extracellular domain of the receptor on the cell surface to promote MPL activation. In agreement with this hypothesis, L-fucose or D-mannose inhibited ThC-dependent cell proliferation in Ba/F3-HuMpl cells (Fig. 2A). In contrast, the effects of the sugars on TPO-dependent cell proliferation were negligible (Supplementary Fig. 10). The effects of fucose and mannose were concentration-dependent, with half-maximal inhibitory concentration ($IC_{50}$) values of 22.8 and 6460 µM, respectively (Fig. 2B). Furthermore, sugar affinity column experiments showed that ThC bound to fucose and mannose in the presence of $Ca^{2+}$ (Fig. 2C). Isothermal titration calorimetry (ITC) experiments confirmed a dependence on $Ca^{2+}$, and the binding dissociation constants ($K_D$s) of rThC bound to fucose or mannose in the presence of 5 mM $Ca^{2+}$ were 4.72 and $66.2 \times 10$ µM, respectively (Fig. 2D and Table S1). In agreement with the in vitro data, the incorporation of $Ca^{2+}$ in the ThC complex was observed in the crystal structure (see below). Taken together, these data suggested that blockade of ThC binding to MPL sugars inhibits ThC-dependent cell proliferation.

### Structural insights into ThC as a homodimeric lectin

To aid in the determination of the primary structure of ThC and gain further structural insights into ThC, we determined the crystal structure of nThC- and Se-Met-substituted rThC at 1.4-Å resolution (Fig. 3). Both ThCs have a β-sandwich structure composed of nine β-strands. An intramolecular disulfide bridge is formed between Cys3 and Cys111 (Fig. 3A), and two ThC molecules assemble as a homodimer in the crystal (Fig. 3A, B). The structure of Se-Met rThC superposed well onto

**A (i)**

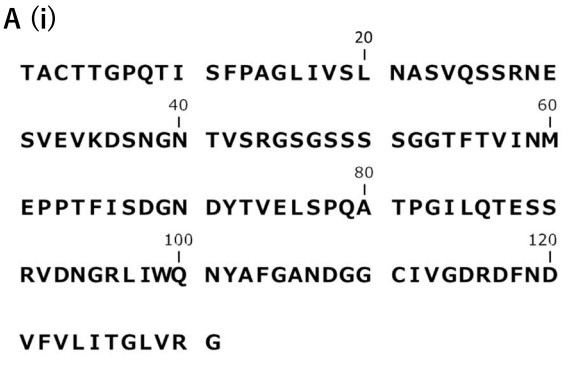

```
            20
            |
TACTTGPQTI SFPAGLIVSL NASVQSSRNE

40                      60
|                       |
SVEVKDSNGN TVSRGSGSSS SGGTFTVINM

            80
            |
EPPTFISDGN DYTVELSPQA TPGILQTESS

100                     120
|                       |
RVDNGRLIWQ NYAFGANDGG CIVGDRDFND

VFVLITGLVR G
```

**(ii)**

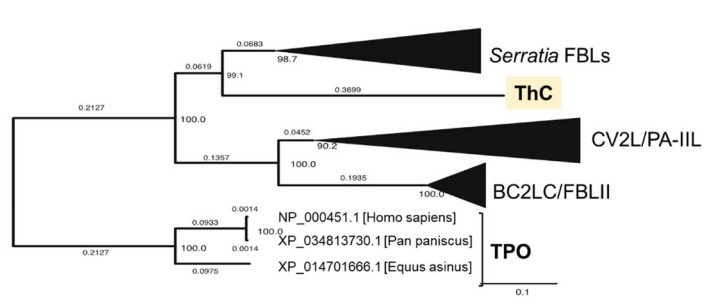

**B**

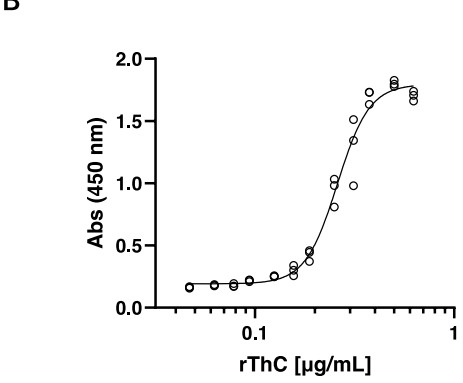

**C**

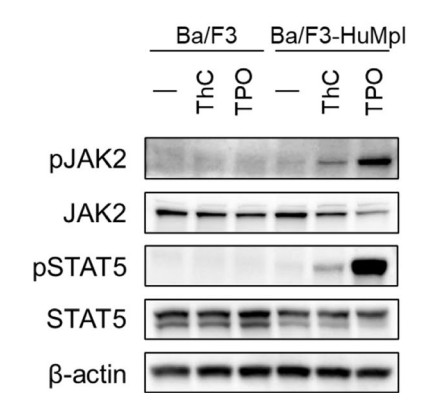

**Fig. 1 | Biochemical profiles of ThC. A** (i) Amino acid sequences of ThC. (ii) Phylogenetic tree of ThC, related bacterial lectins and TPO with collapsed tree nodes (Supplementary Fig. 9). **B** Concentration−response curve of Ba/F3-HuMpl cells proliferating by recombinant ThC (rThC). The half-maximal effective concentration $EC_{50}$ was 0.26 (95% CI of 0.25–0.27) and 0.31 µg/mL (18.6 and 22.1 nM, respectively) for rThC and nThC[12], respectively. **C** Immunoblot analysis of Ba/F3 and Ba/F3-HuMpl cells upon steady-state activation by TPO and rThC. Two independent experiments were performed, and similar results were obtained (see source data file).

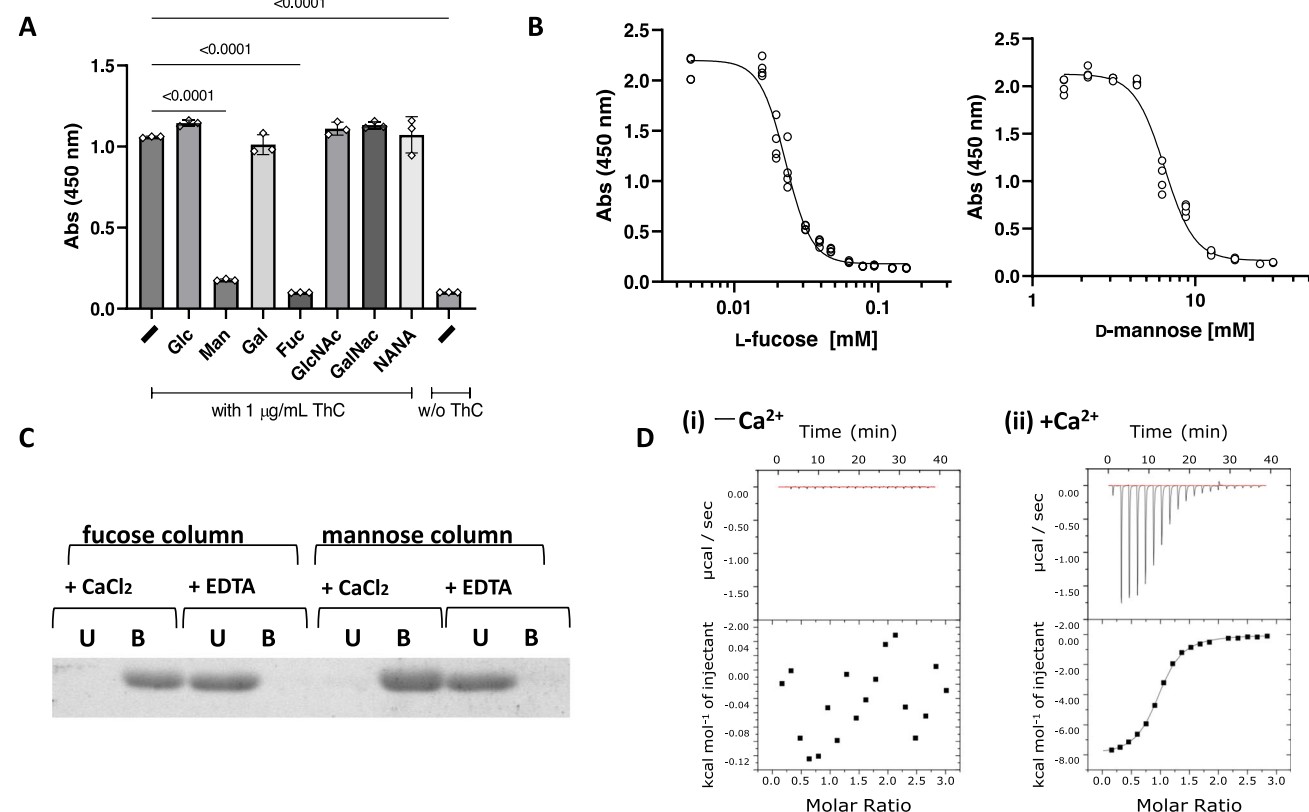

**Fig. 2 | Critical role of sugar-binding capacity in ThC-dependent MPL activation. A** Relative cell proliferation in the presence of various sugars (10 mM) in Ba/F3-HuMpl cells with rThC (1 μg/mL). $n = 3$ per data point, bars ± SD. One-way ANOVA followed by a Dunnett's test for multiple group comparison.
**B** Concentration-dependent inhibition of fucose or mannose on cell proliferation by rThC treatment (1 μg/mL). IC$_{50}$ for fucose and mannose were 22.8 (95% CI of

21.7–23.9), 6460 (95% CI of 6117–6842) μM, respectively. **C** Binding capacity of rThC to fucose- or mannose-immobilizing resins in the presence of 1 mM CaCl$_2$ or 1 mM EDTA. Unbound, U and bound, B. Three independent experiments were carried out and similar results were obtained. **D** Thermodynamic analysis of the interaction with fucose in the absence (left) and presence (right) of 5 mM CaCl$_2$. The thermogram (top) and titration curve (bottom) are shown.

nThC (root-mean-square deviation (r.m.s.d.) of 0.66 Å for 261 Cα atoms), which suggested that it was structurally and functionally equivalent to nThC (Supplementary Fig. 11). Therefore, His-tagged rThC was used for further structural, physicochemical, and physiological analyses and is termed ThC hereafter. Structural insights into ThC, particularly the formation of homodimers, provided a rational model for MPL activation that was likely triggered by the homodimerization of receptor molecules.

**Structural basis for ThC sugar binding**

The crystal structure of ThC in complex with L-fucose (Fig. 3C, Supplementary Note 3) or D-mannose (Supplementary Fig. 12) showed that the sugar molecule was bound in a cavity of the dimeric protein at the interface between two protomers via two Ca$^{2+}$ ions, Ca-1 and Ca-2. Ca-1 was chelated in a polar cavity formed by D117-N119-D120 of one protomer and the C-terminal carboxylate of G131 of the other protomer (denoted as Gly131*, Fig. 3C). This unique structural feature, herein called the pseudodomain-swapping motif, was formed between the protomers and is a characteristic hallmark of this protein family (Fig. 3D). Ca-2 was chelated by polar groups along the N107-D115-D117-D120 sequence (Fig. 3C). Sugar is recognized by ThC via polar interactions along the Ca-1-Ca-2-D115-D120-D108 sequence of one protomer and three hydroxy groups of the carbohydrates. Notably, the pseudodomain-swapping structure enables stable carbohydrate binding via a hydrogen bond network established between the carboxylate of G131* of the adjacent protomer and Ca-1 and O4 of fucose (O2 of mannose). The positions of the carbohydrates were further stabilized by binding between O5 and Ser27 in both sugars. This

manner of recognition shows the importance of Ca$^{2+}$ in carbohydrate binding, which is consistent with the Ca$^{2+}$ dependency of carbohydrate binding of ThC (Fig. 2C, D and Supplementary Table 1). The stereochemical orientation of the three hydroxyl groups is shared between L-fucose and D-mannose but not the other chain-bearing sugars tested, which explains the carbohydrate specificity of ThC (Supplementary Fig. 13). A 1:1 stoichiometry between ThC and mannose/fucose was apparent in the ITC analysis (Supplementary Fig. 14).

In the sequencing study of ThC, we coincidentally found that Q25 was a key residue for its agonist action (Supplementary Note 1); i.e., Q25K did not promote the proliferation of MPL activation-dependent Ba/F3-HuMpl, even at high concentrations (Supplementary Fig. 3b). The ITC data showed a complete lack of affinity of Q25K for fucose (Supplementary Fig. 14). To examine its structural basis, the Q25K mutant was crystallized in the presence of Ca$^{2+}$. The C-terminus in Q25K faced away from Ca-1. The structure of Q25K clearly differed from nThC in the conformation of the C-terminus in the counterpart protomer and the position of the side chain of Q25 (Fig. 3E). This conformational change caused the loss of the pseudointerprotomer domain swapping and resulted in loss of Ca-1 coordination of the C-terminal carboxylate group of G131*, leading to the profound loss of fucose-binding capability and agonist activity of the mutant (Supplementary Figs. 3B, 14). We confirmed this series of changes by preparing a G132 mutant in which an extra G residue was added to the C-terminus to alter the pseudodomain-swapping motif. As expected, ITC analysis revealed that the G132 mutation exhibited diminished fucose-binding activity (Supplementary Table 1 and Supplementary Fig. 14), and agonist activity was completely lost (Supplementary Fig. 15). These

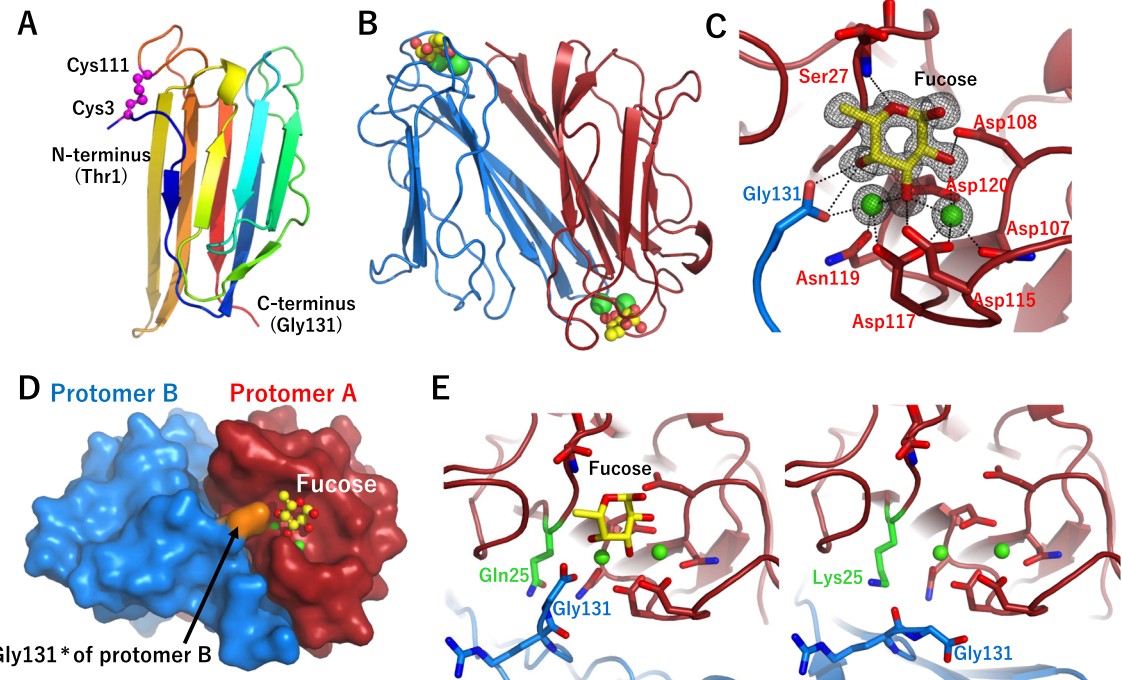

**Fig. 3 | Crystal structure of ThC. A** Ribbon diagram of the nThC monomer colored according to the sequence in blue at the N-terminus to red at the C-terminus. The disulfide bond between Cys3 and Cys111 is shown as a magenta ball. **B** Dimer structure of rThC in complex with $Ca^{2+}$ (green ball) and fucose (ball-and-stick model, yellow: carbon, red: oxygen). **C** A close-up view of the fucose-binding site of rThC. The bound $Ca^{2+}$ ions and fucose are shown as green balls and stick models, respectively. Residues are colored according to the chain as in (**B**). A Fo-Fc map of fucose and $Ca^{2+}$ contoured at 3.0 σ is shown. **D** Pseudodomain swapping structure in dimeric rThC. Gly 131*, shown in orange, of one protomer intervenes in the other. **E** Structural comparison of the $Ca^{2+}$-binding configuration between wild type (left) and Q25K (right). A close-up view of the fucose-binding site is shown. Individual protomers are shown in red and blue. The substituted residues (Q25 and K25) are shown in green.

observations led to the identification of the structural determinants for the sugar-binding and agonist actions of ThC. Specifically, the binding cavity of one protomer, two calcium ions, and the C-terminal domain of the other protomer together stabilize the sugar-bound state of the protein.

**Activation of MPL by ThC via a fucosylated sugar chain**

To gain further insight into the sugar-mediated activation of MPL, we assessed the effects of lectins bearing fucose- or mannose-binding properties on the proliferation of Ba/F3-HuMpl cells. Although the general structural features represented by a β-strand-rich pseudodomain swapping homodimeric structure are common in all the lectins tested, none, except PA-IIL[13,14], a fucose-binding lectin homologous to ThC, promoted the proliferation of Ba/F3-HuMpl cells (Fig. 4A). Despite the high degree of structural similarity between ThC and PA-IIL (r.m.s.d. 2.05 Å for 104 Cα atoms, Supplementary Fig. 16), PA-IIL showed approximately 70-fold reduced potency in inducing MPL-dependent cell proliferation compared with ThC (Fig. 4B). We thus compared the crystal structures of rThC and PA-IIL with or without sugars since some differences in thermodynamic profiles between two lectins in ITC experiments suggested discrete modes of sugar bindings (Supplementary Table 1). However, no obvious differences that may pose their different agonist actions were found (Supplementary Note 2).

This result suggested that although certain structural features inherent to ThC and PA-IIL contributed to their agonist action, the dimerization and sugar specificity of a lectin alone are insufficient for an MPL agonist. The lack of agonist activity found for BC2L-C-CTD, which belongs to the same family as PA-IIL (r.m.s.d. of 1.18 and 2.13 Å for 111 and 106 Cα atoms with PA-IIL and ThC, respectively), strongly suggested that sugar-binding properties, specifically fucose binding,

are crucial for MPL activation in addition to structural similarity (Supplementary Fig. 16). The ITC data of PA-IIL and BC2L-C-CTD clearly showed that PA-IIL bound to fucose and mannose[15], and BC2L-C-CTD bound only to mannose (Supplementary Fig. 17). These data support the importance of fucose-binding activity for MPL activation. Notably, although the fucose-binding affinity of PA-IIL was stronger than that of ThC (Supplementary Table 1), MPL activation by PA-IIL was markedly weaker than that by ThC (Fig. 4B). Therefore, we determined the inherent structural differences between the lectins by comparing the positions of two fucose molecules bound to ThC and PA-IIL. The distance between the representative atom of fucose O4 was 39.6 Å for ThC and 36.4 Å for PA-IIL (Fig. 4C). When one fucose molecule was superimposed, the position of the other was shifted by approximately 6.7 Å, such that the angle between the three O4 atoms was 8.9 deg (Fig. 4C).

To demonstrate the importance of the fucosylated sugar chain, we treated Ba/F3-HuMpl cells with peracetylated 6-alkynyl fucose (6-Alk-Fuc), an inhibitor of GDP-fucose synthase/TSTA3, which attenuates the formation of fucose-containing sugar chains[16]. The resulting 6-Alk-Fuc Ba/F3-HuMpl cells were treated with ThC or TPO. 6-Alk-Fuc decreased ThC- and TPO-induced cell proliferation, with $IC_{50}$ values of 1.2 and 4.8 μM, respectively (Fig. 4D). These results indicated the considerable contribution of fucosylated glycans to receptor activation. However, the type of fucosylated chain interacting with ThC could not be specified via an analysis of the aforementioned data alone. Human FUTs catalyze α(1,2)-, α(1,3)-, α(1,4)-, α(1,6)-, and O-fucosylation, and cell surface glycans may exhibit any of these fucosylation patterns[17]. Therefore, we tested the effect of hypnin, an algal lectin with highly strict recognition of core α(1,6)-fucosylated glycans[18], on the action of ThC. We found that hypnin inhibited ThC-induced cell proliferation in a concentration-dependent manner, with

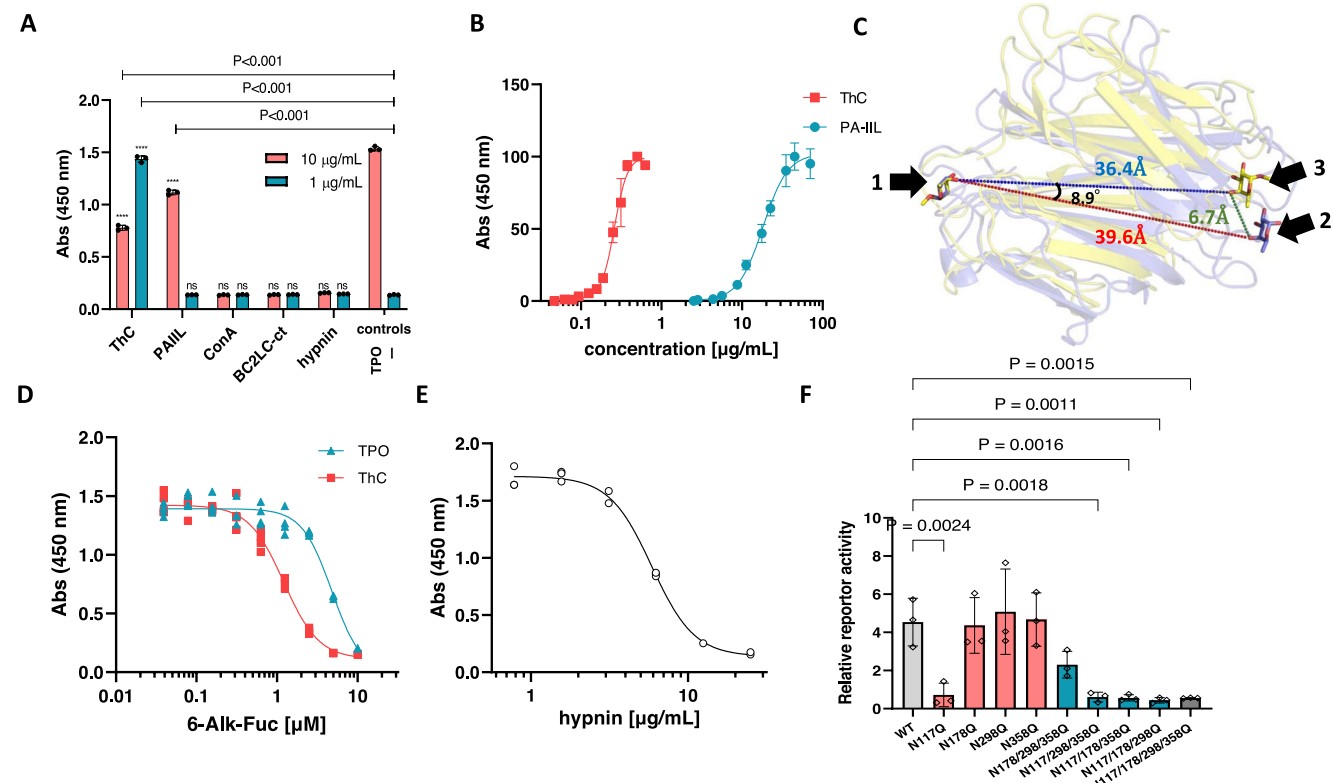

**Fig. 4 | Involvement of the fucose moiety in ThC-dependent MPL activation.**
**A** Effect of lectins on the proliferation of Ba/F3-HuMpl cells: PA-IIL, a fucose/man-nose selective bacterial lectin with high homology to ThC; ConA, a mannose-specific legume lectin; BC2L-C-CTD, mannose-selective bacterial lectin with high homology to ThC; hypnin, a core 1,6-fucosylated glycan-specific algal lectin. TPO, 10 ng/mL; (-), no agonist. Note that the cell proliferation was suppressed in the presence of 10 μg/mL ThC presumably due to the suppression of cell proliferation by its agglutinating activity and to the interference of receptor dimerization by occupying a ligand-binding site on a receptor molecule. **B** Concentration dependency of Ba/F3-HuMpl cell proliferation induced by PA-IIL (EC$_{50}$ = 18.1 μg/mL, 95% CI of 16.7–19.7) and rThC. **C** Relative positions of two fucose molecules bound to the ThC dimer and PA-IIL dimer. One of the two fucose molecules bound to each is superimposed (fucose on the left). The other fucose molecule is shown as sticks. Purple represents fucose bound to ThC, and yellow represents fucose bound to PA-IIL. The numbers represent the distance between O4 atoms (red: between two

fucoses of ThC, blue: between two fucoses of PA-IIL, green: between fucoses of ThC and PA-IIL) and the angle between the three O4s of superimposed fucoses, of the PA-IIL-bound fucose, and the ThC-bound fucose. Ribbon diagrams of the ThC dimer (purple) and PA-IIL dimer (yellow) are also shown in translucent form. **D** Preferential inhibition of ThC-dependent cell proliferation in Ba/F3-HuMpl cells and 6-alkynyl-fucose. Cells were cultured in the presence of ThC (1 μg/mL) or TPO (10 ng/mL). IC$_{50}$ values for TPO and ThC were 4.8 (95% CI of 4.0–8.5) and 1.2 (95% CI of 1.1–1.3) μM, respectively. **E** Hypnin-mediated inhibition of cell proliferation induced by ThC (1 μg/mL), with an IC$_{50}$ value of 5.9 (95% CI of 5.5–6.2) μg/mL. **F** Effect of potential N-glycosylation site mutations on MPL for activation by ThC (1 μg/mL). STAT5 reporter activity representing MPL activation status is depicted. Data were presented as mean value ± SD, $n$ = 3. Gray bar: wild-type (WT) MPL; red bars: single-site mutant MPL; blue bars: triple-site mutant MPL; and white bar: quadruple-site mutant MPL. One-way ANOVA followed by a Dunnett's test for multiple group comparison.

an IC$_{50}$ of 5.9 μg/mL. Because the cytostatic concentration of hypnin was markedly higher (approximately 47 μg/mL, Supplementary Fig. 18), this result was ascribed to competitive inhibition between ThC and hypnin for core α(1,6)-fucose (Fig. 4E).

To further examine the molecular basis of sugar-mediated MPL activation, we assessed the actions of ThC against glycan mutants of MPL. The extracellular domains of MPL have four consensus amino acid sequences for N-type glycans at N117, N178, N298, and N358[19]. To determine the critical site for ThC activation, mutants in which N residues were replaced by Q residues were expressed in HEK293T cells, and receptor activation was monitored using the STAT5 reporter assay. The mutations did not significantly affect receptor activation by TPO (Supplementary Fig. 19), showing that the mutation itself had little effect on the receptor in terms of cell surface expression and activation. In contrast, the N117Q mutant completely lost sensitivity to ThC (Fig. 4F). In a reciprocal experiment, in the N178/298/358Q mutant MPL, where only the N117 consensus site remained, the MPL responded to ThC, whereas the other mutant MPL, in which three of the four consensus N residues were replaced with Q residues, was inert to ThC (Fig. 4F). These data, together with the

observation that the lectin property of ThC is critical for MPL activation, implied that MPL was activated via ligand binding at the glycan attached to the N117 residue of MPL on the cell surface.

Notably, the glycan site identified here is the same site that was previously attributed to activation by CALRmut under pathological conditions[5–7,20]. Because the property of CALRmut for MPL remains largely elusive due to a lack of an assay system (see "Introduction"), we examined the mode of ThC-induced receptor activation. Unlike TPO, which induced the activation of MPL in 10 min and resulted in a rapid attenuation of activation, ThC activated MPL at 30 min after ligand addition and exhibited a capacity for prolonged activation (Fig. 5A). When the levels of accumulated cell surface receptors were measured upon activation with a set of agonists, the receptors remained on the cell surface in the ThC-treated cells (Fig. 5B). This finding differed from the receptors in TPO-treated cells because the receptor population gradually decreased due to internalization (Fig. 5B). Similarly, prolonged accumulation of MPL on the cell surface was observed in CALRmut-expressing cells, where receptor activation was persistent[7]. These observations suggest that the mode of activation by lectin-type ligands is slow and steady, but cytokine-mediated activation is rapid and extinctive.

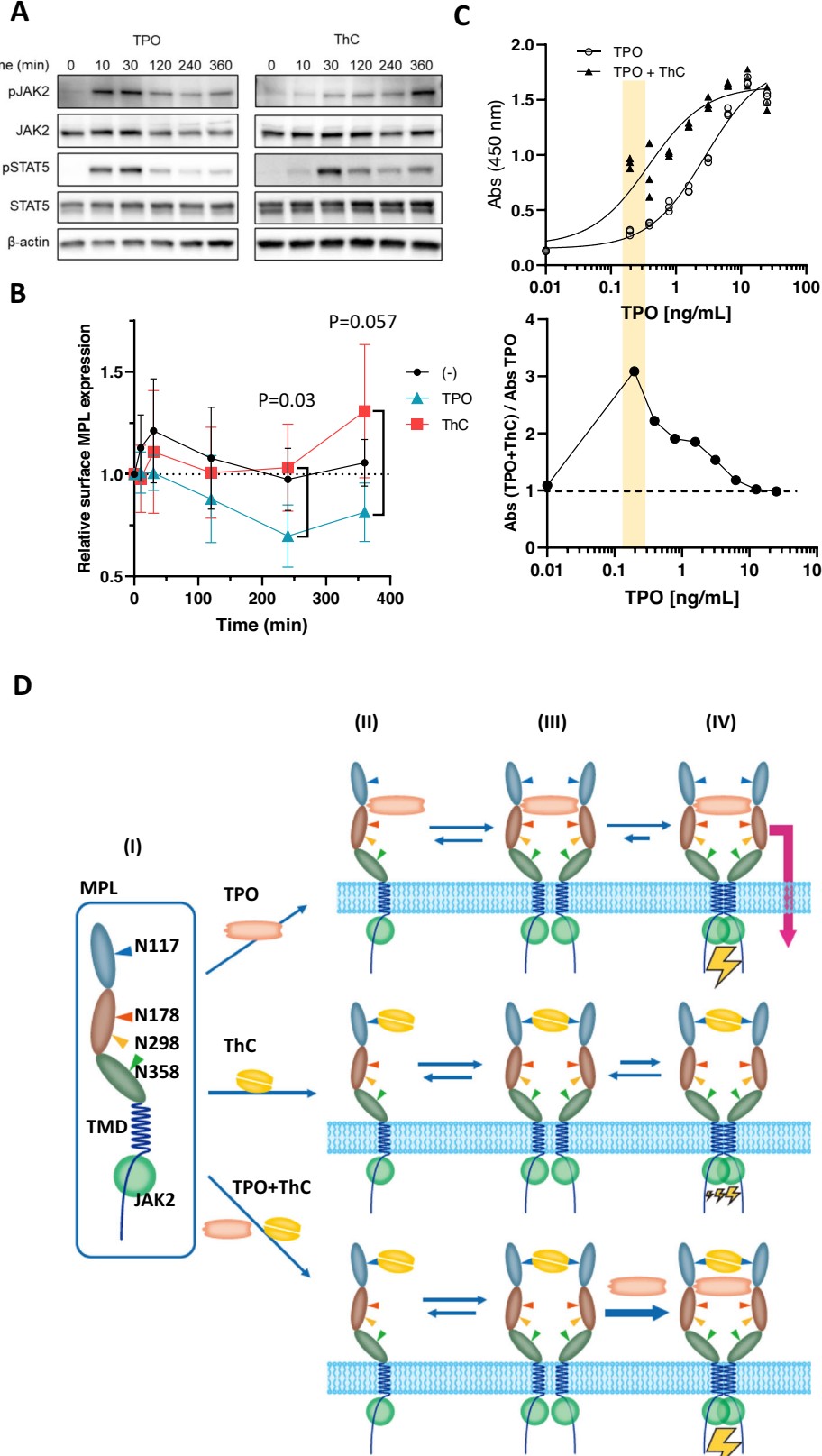

## Potentiation of TPO activity by lectins

Our observations strongly support the hypothesis that MPL can be activated by a completely different mechanism from that of currently known agonists. Because lectin-type ligands likely maintain increased levels of receptors on the cell surface, we assessed whether these ligands synergized with natural ligands. We examined the concentration-dependent proliferation of TPO in Ba/F3-HuMpl cells in the presence of a subactivating concentration of ThC (0.1 μg/mL). As expected, ThC synergistically enhanced the agonist action of TPO (Fig. 5C). The rate of enhancement had a bell-shaped relationship with TPO concentration and showed the greatest effect when 0.2 ng/mL TPO was applied (Fig. 5C, lower tracing). At this concentration, TPO

**Fig. 5 | Differential processes of MPL activation. A** Time-dependent activation of MPL-downstream molecules in Ba/F3-HuMpl cells treated with ThC (1 μg/mL) and TPO (3 ng/mL). Phosphorylation of STAT5 and AKT was monitored at the indicated times after the addition of an agonist. Three independent experiments were performed, and similar results were obtained (see source data file). **B** Relative amount of MPL on the cell surface 0, 10, 30, 120, 240, and 360 min after the addition of ThC (1 μg/mL) and TPO (3 ng/mL) was measured with three independent experiments, bars indicate mean ± SD. Relative surface MPL at 240 and 360 min between ThC and TPO differed significantly, with $P = 0.030$ and $0.057$, respectively, two-tailed paired $t$-test. For immunoblot data, see Supplementary Fig. 20. **C** Synergistic effects observed in each combination of agonists in the Ba/F3-HuMpl cell proliferation assay. Concentration–response curves for tested agonists in the absence or presence of fixed subactivation concentrations of ThC (0.1 μg/mL) (upper trace). The ratios of absorptions with and without ThC (0.1 μg/mL) are plotted (lower trace). $EC_{50}$ values for, with and without ThC are, 0.4 (95% CI of 0.2–0.7) and 2.8 (95% CI of 2.1–3.7) ng/mL, respectively. Non-liner fit variable slope with three parameters were used. **D** Proposed mechanisms of activation by two mechanistically discrete agonists, TPO and ThC: (I) Schematic depiction of MPL. Monomeric MPL has four N-glycosylation sites at N117, 178, 298, and 358; (II) ligand-bound monomeric state with each TPO and ThC; (III) ligand-bound nonactivated state; and (IV) dimeric signaling complexes.

alone induced only a 10% increase in cell proliferation but reached 50% in the presence of ThC. The $EC_{50}$ value of TPO with ThC (0.37 ng/mL, 20 pM) was 7.5 times lower than that of TPO alone (2.8 ng/mL, 150 pM). These data supported the idea that bidentate glycan-binding ligands allosterically sensitized the action of TPO.

## Discussion

We determined the three-dimensional structure of the potent MPL agonist ThC as a homodimeric complex of lectin molecules featuring two calcium ions that form a cage-like complex for selective binding to fucose and mannose (Figs. 2C, 3B, C). We showed that ThC bound to fucose and mannose to a lesser extent (Supplementary Table 1) and demonstrated that the fucose-binding property was critical for MPL activation (Fig. 4A, D, E). The mode of MPL activation by ThC resembled the pathogenic ligand CALRmut because of the dependency of the N-glycosylation site on activation (Fig. 4F) and the persistent accumulation of receptor molecules on the cell surface (Fig. 5B). The lectin-type agonist promoted the surface accumulation of cytokine receptors by blocking internalization (Fig. 5B, Supplementary Fig. 21) and induced a slow but steady activation of downstream molecules (Fig. 5A) that synergized with natural ligands (Fig. 5C). These data elucidate the previously understudied molecular mechanism of cytokine receptor activation by lectins.

This study presents the evidence that an exogenous ligand activates MPL via surface glycans on the receptor. Notably, the glycan critical for MPL action is the same one required for activation by the internal ligand CALRmut[5–7]. To validate the potential of ThC as an agonist for MPL, we performed in vitro assay with human hematopoietic stem cells derived from induced pluripotent stem cells and examined the potency of ThC in megakaryocytic differentiation. However, the assay was perturbed by the cell aggregation induced by ThC that harbors an agglutinating activity. We even analyzed the aggregated cells, but no obvious differentiation of megakaryocytes was observed in the presence of ThC (Supplementary Fig. 22). Although in vivo efficacy of ThC in mouse models is of great interest, the modification of ThC to reduce its adverse actions is necessary for the experiment.

In cells expressing CALRmut, homomultimerized CALRmut engages with MPL bearing immature N-glycans at N117 in the endoplasmic reticulum (ER) to form a 2 + 2 quadripartite complex of MPL-CALRmut in the Golgi apparatus, and this complex is trafficked to the cell surface for activation[6,7,21]. CALRmut fails to activate MPL expressed on the surface of cells that do not express CALRmut[10,11], which renders the potential of CALRmut as the MPL ligand uncertain and leaves the molecular mechanism of MPL activation ambiguous. The present study clearly demonstrated that the N-glycan at N117 is the bona fide switch for the activation of MPL by homodimerized lectins and ensures the agonistic effect of homomulitimerized CALRmut on MPL.

Our structural insights into ThC led to the discovery of PA-IIL as a ThC-type MPL agonist, which demonstrates the potential of bacterial fucose-binding lectins, including theoretical lectins, as MPL agonists. Structural and biological comparisons of ThC and PA-IIL suggested that the dimerized form and capacity of fucose binding

were not sufficient for MPL activation. The position of MPL molecules, which is determined by the positioning of the fucose moiety at N117 of MPL bound by the lectin, plays a crucial role in the degree of receptor activation. Notably, we found that the spatial arrangement of two fucose-binding pockets clearly differed between ThC and PA-IIL. When the core-(1,6) fucose of the sugar chain at N117 of each MPL bound tightly to two ligand-binding cores of the lectin, the configuration of the lectin-bound receptor complex directly reflected the spatial relationship of the pocket. Therefore, ThC- and PA-IIL-bound activating receptor complexes differ structurally. We propose that this structural difference affects the efficacies of the two lectins. A recent study demonstrated that dimeric antibodies that bridge and dimerize MPL by binding various sites near the canonical ligand-binding domain of MPL activated the receptor in distinctive manners, which resulted in agonist-based decoupling of HSC self-renewal and differentiation[22]. Because no structural information on the active receptor complex for MPL was known, the mechanistic basis of this phenomenon was elusive. However, our observations, in conjunction with the above study, support the hypothesis that slight structural differences in the extracellular domain of MPL affect receptor activation and signaling.

Our results showed that the ThC-dependent activation of MPL persisted and was associated with sustained expression of MPL on the cell surface. These results were also observed in CALRmut-dependent MPL activation[7], suggesting a conserved mechanism of action in lectin-mediated receptor activation. The activation dynamics of MPL are largely controlled by the internalization and recycling of the receptor and the de novo biosynthesis of new receptors[23]. Our results suggested that internalization of ThC-activated receptors was a slow process, as observed in CALRmut-activated receptors, which yielded persistent activation. The strong synergy observed in the ThC/TPO coapplication may be partially ascribed to the sustained signaling of long-lasting cell surface receptors (Supplementary Fig. 21).

MPL belongs to the class I cytokine receptor family, whose activation relies on the formation of receptor dimers. However, the process of dimer formation is not well understood[24]. Preformed dimers are likely activated upon conformational alterations[24,25]. However, recent single-molecule live cell imaging of MPL expressed on HeLa cells revealed that the population of monomeric receptors surpassed the predimeric form, and the monomer assembled into dimers after interacting with agonists[26]. Because we observed weak but sustained receptor activation by ThC associated with sustained cell surface expression of the receptor, we propose that a transition state, a ligand-bound but subtle-activated dimer in the process of signaling dimer formation, plays an important role (Fig. 5D III). The role of this well-conceivable transition complex formed in the middle of the activation process was previously hidden because this intermediate is short-lived in TPO-activated MPL due to potent native interactions between the receptor and ligand[26]. The strong synergy of ThC with TPO supports the presence of this transition state because predimerization reduces activation barrier existing in between the state II and III (Fig. 5D). Therefore, ThC stabilizes transition state III, although ThC-bound III eventually shifts to form signaling complex IV, presumably via the aid

of intrinsic domain interactions in the transmembrane (TMD) and intracellular domains (Fig. 5D).

Because exogenously applied secreted CALRmut fails to activate normal MPL[10,11], ThC- and ThC-type fucose-binding lectins were the only probes, and they were excellent for studying the receptor kinetics and dynamics of MPL during N-glycan-mediated activation. The present study reports the structural basis of the sugar-mediated activation of cytokine receptors. MPL-mediated signaling is involved in at least two discrete activation processes in hematopoiesis, hematopoietic progenitor cell differentiation/megakaryocyte formation and HSC self-renewal/maintenance[1–3]. We propose that fucose-binding lectins are novel tools to control structure to activate the MPL complex and enable fine-tuning of dimerization, internalization, and signaling in conjunction with coapplication with other agonists.

## Methods

### Ethics statement
The use of iPS cells was conducted in accordance with the Declaration of Helsinki and approved by the ethics committee of Juntendo University School of Medicine (IRB#M12-0895).

### Reagents
Anti-MPL (Merck Millipore #06-044, dilution 1:2000), anti-STAT5 (Cell Signaling Technology #94205, dilution 1:2000), anti-phospho-STAT5 (Cell Signaling #9359, dilution 1:1000), anti-JAK2 (Cell Signaling #3230, dilution 1:2000), anti-phospho-JAK2 (Cell Signaling #3771, dilution 1:1000), anti-β-Actin (Cell Signaling Technology #4967, dilution 1:20,000), anti-phospho-ERK1/2 (Cell Signaling Technology #9101, dilution 1:1000), anti-phospho-AKT (Cell Signaling Technology #4060, dilution 1:1000), anti-STAT5 (Cell Signaling Technology #94205, dilution 1:2000), anti-ERK1/2 (Cell Signaling Technology #9102, dilution 1:2000), anti-AKT (Cell Signaling Technology #9271, dilution 1:2000), and recombinant human TPO (PeproTech #300-18 for Fig. 1B, Fig. 2A, Fig. 4A, D, Fig. 5C, Kyowa Hakko Kirin for Fig. 1C, Fig. 5A, B) were used.

### Affinity purification of native ThC
The sponge specimen used here was collected in Chuuk State of Federated States of Micronesia in 2009 under permission of Department of Marine Resources, Chuuk State FSM, and was extracted as described previously[12]. A sponge aqueous extract was treated with acidic buffer (pH 3.0) to obtain a ThC-enriched extract. The extract was applied to a 1-mL Sepharose-fucose affinity gel (EY Laboratories, Inc.). The column was eluted first with 50 mM Tris-HCl buffer and then with fucose. The fucose eluent was dialyzed to yield purified protein.

### Cell culture and proliferation assay
ThC cell proliferation assays were performed as described previously[12]. Briefly, the murine interleukin-3-dependent pro-B-cell line Ba/F3 expressing human MPL (Ba/F3-HuMpl cells)[27] was precultured for 4 days and then harvested by centrifugation at $160 \times g$ for 3 min. After washing with PBS (-), the collected cells were resuspended in RPMI-1640 medium containing 10% FBS at a cell density of $6.0 \times 10^4$ cells/mL. A 90-μL aliquot of the cell resuspension was transferred to a 96-well plate. In the presence of various concentrations of ThC or recombinant ThCs (10 μL), the cells were cultivated at 37 °C under air with 5% $CO_2$. PBS and TPO were used as negative and positive controls, respectively. After 4 days of cultivation, cell proliferation was measured with a cell counting kit (Dojindo). A 10-μL aliquot of the cell counting kit was added to each well. After incubation for 2 h, the absorption at 450 nm (Abs450) was recorded with a microplate reader. For the 6-alkynyl-fucose assay, Ba/F3-HuMpl cells were pretreated with 6-Alk-Fuc for 6 h, ThC or TPO was added, and the cells were cultured for 4 days.

### Preparation of samples for the cell proliferation assay
ConA (Sigma), hypnin (from *Hypnea japonica*), PA-IIL (Fujifilm-Wako), and BC2LC-CTD (recombinant) were dissolved in PBS (-). 6-Alkynyl fucose (Peptide Institute, Inc.) was dissolved at 100 mM in DMSO and diluted with PBS (-) to each concentration. Sugar solutions were prepared with PBS (-) except N-acetylneuraminic acid (NANA). NANA was suspended in PBS (-) and neutralized with aqueous NaOH to pH 7.0. All the reagents were filter-sterilized with a 0.2-μm filter prior to use.

### Determination of the amino acid sequence
Draft amino acid Edman degradation was performed using ThC purified by SDS−PAGE. The gel was electroblotted on a PVDF membrane, and the band for ThC was cut out for the N-terminal amino acid sequence using an automated sequencer (Procise 492HT). The internal amino acid sequence was obtained by digesting purified ThC with either trypsin (Fujifilm-Wako), chymotrypsin (Fujifilm-Wako) or V8 protease (Fujifilm-Wako). Each digest was separated by HPLC using a reversed-phase column (VYDAC protein&peptide C18) with a gradient (0–50%) of 0.1% aqueous TFA and acetonitrile. Each of the peptide fragments was subjected to de novo sequence analysis using MALDI-TOF MS/MS and to Edman degradation. The deduced amino acid sequences were mapped to give a draft amino acid sequence of ThC. The entire amino acid sequence was then confirmed by mass spectrometry as follows. A drop (4.5 μL) of crystallization supernatant from the X-ray analysis experiment containing 7.0 μg of native ThC was mixed with 80 μL of acetone and then centrifuged at $19,000 \times g$ and 4 °C for 15 min. The precipitate was resuspended in 80 μL of acetone. After centrifugation at $19,000 \times g$ and 4 °C for 15 min, the precipitate was further washed as described above and then air-dried. The dried sample was resuspended in 40 μL of 1× phase transfer surfactant (PTS)[28]. A 20-μL aliquot of resuspended sample was incubated with 2 μL of 200 mM Bond-Breaker TCEP solution (Thermo Fisher Scientific) to cleave the disulfide bond for 30 min at 50 °C. The reduced thiol was alkylated by 2 μL of 375 mM 2-iodoacetamide for 30 min at room temperature in the dark. After alkylation, an excess amount of 2-iodoacetamide was reacted with 2 μL of 400 mM L-Cys for 10 min at room temperature in the dark. The alkylated sample was digested by either 200 ng of trypsin (Promega) with 200 ng of Lys-C (Fujifilm-Wako) at 37 °C overnight or 200 ng of chymotrypsin (Promega) with 10 mM $CaCl_2$ at 25 °C overnight. A total of 30 μL of the digestion sample was precipitated by the addition of 45 μL of 1.7% trifluoroacetic acid (TFA). After centrifugation at $19,000 \times g$ and 4 °C for 15 min, the supernatant was purified using a Stage-Tip as described previously[29]. Peptide fragments were eluted from the Stage-Tip using 70% acetonitrile and 0.1% TFA, and the elution was freeze-dried. Recombinant ThC (rThC Q25, 2.2 μg) dissolved in 20 μL of PTS was also digested according to the protocol described above. The peptide fragments were dissolved with 10 μL of 0.1% TFA, and the peptides of the native ThC and rThC Q25 were analyzed with a quadrupole Orbitrap benchtop mass spectrometer (Q-Exactive, Thermo Fisher Scientific) equipped with a Nanospace SI-2 HPLC system (Osaka Soda Co., Ltd). The column temperature was maintained at 45 °C. The flow rate of the mobile phase was 200 μL/min; mobile phase A consisted of 0.05% formic acid (FA), and mobile phase B consisted of 0.05% FA/90% acetonitrile. The mobile phase gradient was programmed as follows: 0% B (0–2 min), 0–35% B (2–12 min), 35–55% B (12–15 min), 55–80% B (15–16 min), 80% B (16–18 min), 80–0% B (18–18.5 min), and 0% B (18.5–20 min). MS data acquisition was performed using Xcalibur 3.0.63 (Thermo Fisher Scientific). MS1 spectra were collected in the scan range of 350–1200 $m/z$ at 70,000 resolution and 200 $m/z$ to hit an AGC target of $1 \times 10^6$ with an injection time of 200 ms. The AGC target value for fragment spectra was set to $1 \times 10^5$, and the intensity threshold was maintained at $3.3 \times 10^4$. The isolation width was set to 2.4 $m/z$, and the 12 most intense ions were fragmented in a data-dependent mode by collision-induced dissociation with a normalized collision energy of 27.

The amino acid sequences of the draft sequence and rThC Q25 were added to the UniProt sequence database (release 31st July 2019,

entry 557,016, all species, reviewed). The MS file was searched against the database using Proteome Discoverer 1.4.0.288 (Thermo Fisher Scientific) and PEAKS Studio 10.0 Build 20190129 (Bioinformatics Solutions). The setting parameters were as follows: enzyme, trypsin (semi) or chymotrypsin (semi); maximum missed cleavage sites, 2 (Proteome Discoverer) or 4 (PEAKS); precursor mass tolerance, 6 ppm; fragment mass tolerance, 0.02 Da; fixed modification, cysteine carbamidomethylation. The peptide identification was filtered to a false discovery rate of less than 1%.

## Overexpression and purification of recombinant ThC (rThC), rThC mutants, and BC2LC-CTD

An expression vector of Q25K mutant and BC2LC-CTD was constructed by cloning a synthesized DNA fragment corresponding to the amino acid sequence of ThC Q25K determined by mass spectrometry and the C-terminal domain of BC2LC into the NdeI/XhoI site of a modified pET28 vector. The 6× His-tag was attached to the N-terminus of ThC because crystal structure demonstrated that N-terminus of ThC and BC2LC is located far from the sugar-binding site, and therefore His-tag attached to N-terminus does not affect the fucose binding. The expression vectors of wild-type rThC and G132 mutant were constructed by inverse PCR with PrimeSTAR Max DNA Polymerase (Takara Bio) using primers shown in Supplementary Table 3 and the expression vector of Q25K mutant and rThC wild type as a template, respectively.

*Escherichia coli* strain BL21(DE3) harboring the expression vector of the desired protein was cultivated in LB medium at 37 °C with shaking at 120 rpm. When the OD600 reached 0.6, isopropyl-β-D-thiogalactopyranoside (IPTG) was added to the medium at a final concentration of 0.2 mM to induce the expression of rThC, and the mixture was then incubated at 25 °C overnight.

Cells were harvested by centrifugation at $4000 \times g$ for 30 min. The collected cells were suspended in buffer A composed of 20 mM HEPES-NaOH (pH 8.0) and 200 mM NaCl and then disrupted with a UD-211 ultrasonic disruptor (TOMY SEIKO). After centrifugation at $40,000 \times g$ for 30 min, the supernatant was loaded onto a 1-mL column of Ni Sepharose (GE Healthcare). After washing with sonication buffer, the bound protein was eluted using a concentration gradient of imidazole in the sonication buffer. Fractions containing purified rThC were further purified by size-exclusion chromatography using HiLoad 26/600 Superdex 75 pg (GE Healthcare) preequilibrated with buffer A. Fractions containing rThC were collected and used for further experiments. SeMet-substituted rThC was expressed and purified by the same method as rThC with the exception that SeMet-substituted M9 medium was used instead of LB medium.

## Isothermal titration calorimetry (ITC)

ITC measurement was performed with an iTC200 (GE Healthcare) in 20 mM HEPES-NaOH (pH 8.0), 200 mM NaCl at 25 °C. The cell was filled with approximately 100 μM rThC, 100 μM BC2LC-CTD, or 35 μM PA-IIL, and the syringe was filled with 1.5 μM fucose or mannose. The ligand was injected 18 times in a portion of 2 μL over 120 s. The data were analyzed with the program ORIGIN7 SR4 [v7.0552 (B552)].

## Carbohydrate binding assay

The carbohydrate-binding specificity of rThC was analyzed with a carbohydrate Gel Kit#1 (EY Laboratories). The binding to fucose, mannose, lactose, N-acetylglucosamine, and N-acetylgalactosamine was evaluated with resin in which each carbohydrate was immobilized. A 0.5-mL aliquot of 0.1 mg/mL purified rThC was loaded on 0.1 mL of resin immobilizing each carbohydrate. After washing with 0.5 mL of buffer, the bound rThC was eluted by elution buffer containing 0.2 M fucose. The specific binding of rThC to the carbohydrate was evaluated by SDS−PAGE.

## Crystallization, X-ray diffraction data collection, and structure determination

For crystallization, purified proteins concentrated up to approx. 6 mg/mL were used. Crystallization was carried out by sitting-drop vapor diffusion method at 20 °C. Crystals of nThC were grown from a buffer composed of 0.1 M Tris-HCl (pH 8.5), 0.2 M MgCl₂, 30% (w/v) PEG 4000. The diffraction dataset of the nThC was collected at Advanced Photon Source (IL, USA). The diffraction data of nThC were processed with the program HKL2000[30].

For phasing of the nThC, the crystal structure of SeMet-substituted rThC (SeMet-rThC) was determined. Crystals of SeMet-rThC were grown from a buffer composed of 100 mM sodium acetate (pH 3.3–5.5), 20% PEG3350-6000, and 20% PEG400. rThC Q25K was crystallized in the presence of 5 mM CaCl₂ because a biochemical analysis revealed that rThC requires Ca²⁺ ions for its carbohydrate-binding activity. Crystals of rThC Q25K in the presence of 5 mM CaCl₂ were grown from a buffer composed of 100 mM sodium acetate (pH 3.3–5.5), 20% PEG3350-6000, and 20% PEG400. Crystals of rThC in complex with fucose or mannose were obtained by cocrystallization, in which 5 mM fucose or mannose was added to the purified rThC solution. X-ray diffraction experiments were conducted in Photon Factory (Tsukuba, Japan) and SPring-8 (Harima, Japan). Diffraction data of SeMet-substituted rThC, rThC in the presence of CaCl₂, rThC in complex with fucose, and rThC in complex with mannose were collected in Photon Factory. The diffraction data of rThC were processed with the program XDS[31]. The statistics of data collection are summarized in Supplementary Table 2.

The crystal structure of SeMet-rThC was determined by the Se-SAD method. The sites of Se were determined using the program HKL2MAP[32]. Phasing and model building were performed using phenix.autosol[33]. The crystal structure of nThC was determined by the molecular replacement method using the program phenix.mr[34] with the structure of SeMet-rThC as the search probe. The crystal structures of rThC and its complexes with Ca²⁺ ions, mannose, and fucose were determined by a molecular replacement method with the structure of nThC as the search probe. An electron density map was calculated using phenix.fft in Phenix program suite (version 1.19.2-4158)[35]. Structure refinement was performed using phenix.refine[36].

## STAT5 reporter assay

To determine the critical site for ThC activation, MPL mutants in which the N residues were replaced by Q residues were expressed in HEK293T cells, and receptor activation was monitored using the STAT5 reporter assay that has been recognized as a model system to validate MPL ligands[21,37,38]. For the expression of MPL with an amino acid substitution from asparagine (N) to glutamine (Q) on a potential N-glycosylation site at N117, 178, 298, and 358, cDNAs were created by PCR mutagenesis (primers listed in Supplementary Table 3) and subcloned into the pcDNA3.1 vector (Life Technologies). V5-tagged mutant MPL and untagged wild-type MPL were used for the reporter assay. All plasmids constructed were verified by sequencing before use. The reporter assay was performed as described previously[21]. Briefly, pcDNA3.1 with MPL cDNA, pGL4.52 (Promega #E4651), and pRL−TK (Promega #E2241) were co-transfected into HEK293T cells using lipofectamine 2000 (Invitrogen #11668019). MPL agonists were added to the media 5 h after the transfection, and then the reporter activity was measured 24 h after the transfection by the Dual-Luciferase Reporter Assay System (Promega #E1910) using a GLO-MAX luminometer (Promega) following the manufacturer's protocol.

## Immunoblot analysis

All immunoblot analyses were performed as described previously[10]. To prepare cell lysates, cells were washed with PBS containing 2 mM orthovanadate and then sonicated in RIPA buffer (20 mM Tris-HCl [pH 7.5], 150 mM NaCl, 1 mM EDTA, 1% NP-40, 1% sodium deoxycholate) containing 2 mM orthovanadate and a protease inhibitor cocktail.

Equal amounts of protein were denatured, electrophoresed, and blotted to polyvinylidene fluoride membranes (Immobilon-P, IPVH00010, Millipore). The blotted membranes were incubated for 1 h at room temperature in TBST buffer (24 mM Tris [pH7.4], 147 mM NaCl, 2.7 mM KCl, 0.1% Tween-20) containing 5% bovine serum albumin (BSA) (034-25462, WAKO) for phospho-specific antibodies or 5% skim milk for other antibodies. After washing membranes with TBST buffer, the membranes were incubated overnight at 4 °C with primary antibodies (described in Reagents) in TBST buffer containing 5% BSA. After washing membrane in TBST buffer, the membranes were incubated with horseradish peroxidase-conjugated goat anti-rabbit IgG (#111-035-003, Jackson Immuno Research) in TBST buffer containing 5% skim milk for 1 h at room temperature. After washing membrane in TBST buffer, the chemiluminescence reaction was performed using SuperSignal West Femto Maximum Sensitivity Substrate (Thermo Scientific), and then images were captured using Fusion FX7 (M&S Instruments Inc.). The data were quantified using ImageJ software.

### Measurement the levels of MPL on cell surface

Ba/F3-HuMpl cells were pre-cultured with the medium in the absence of agonist overnight, and then incubated with the media in the presence or absence of ThC or TPO at 37 °C in a humidified incubator with 5% $CO_2$. For the examination of synergistic effect of ThC and TPO, the pre-culture was performed with the media in the absence of agonist overnight and in the presence of ThC for 2 h before the addition of TPO. Cell surface MPL was isolated using a Cell Surface Protein Isolation kit (Thermo Fisher #89881) and detected by immunoblot analysis as described previously[7] with the following modifications. Briefly, cells were washed with ice-cold PBS (-), reacted with 0.25 mg/mL biotin in PBS (-) for 30 min at 4 °C, quenched with the supplied buffer, washed with ice-cold PBS (-), and sonicated in the RIPA buffer containing 2 mM orthovanadate and a protease inhibitor cocktail. Surface proteins were purified from cell lysates using NeutrAvidin Agarose Resin (supplied in the kit), and then subjected to the immunoblot analysis.

### Statistical analysis

Data were obtained more than triplicate at each data point representing mean value and error bars ±SD. Histograms were analyzed using Dunnett's multiple comparison test. Concentration–response curves are generated using GraphPad Prism 8.0.3. $EC_{50}$ values with 95% confidence intervals of the data fittings with nonlinear regression curve fit with four parameters unless otherwise noted.

### Reporting summary

Further information on research design is available in the Nature Portfolio Reporting Summary linked to this article.

## Data availability

Coordinates and structure factors have been deposited in the Protein Data Bank under accession codes 7F9F (nThC), 7F91 (Met substituted rThC), 7F9G (rThC in complex with $Ca^{2+}$ and fucose), 7FBL (rThC in complex with $Ca^{2+}$ and mannose), and 7F9J (rThC Q25K in complex with $Ca^{2+}$). All other data are available in the main text or the supplementary materials. Amino acid sequence for ThC has been deposited in UniPlot [https://www.uniprot.org/] under accession code C0HM62. Source data are provided with this paper.

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

## Acknowledgements
We thank Professor Yasuhiko Kizuka at Gifu University for providing 6-alkynyl fucose and valuable comments. We also thank Dr. Yinjie Yang and Ms. Mai Nudejima for their technical assistances, Dr. Takanori Nakamura at Nissan Chemical Co. Ltd. and Dr. Megumi Funakoshi-Tago at Faculty of Pharmacy, Keio University for providing Ba/F3 cells, the Global Facility Center at Hokkaido University for performing amino acid sequence analysis, the Laboratory of Molecular and Biochemical Research and the Division of Cell Biology in the Research Support Center of the Juntendo University Graduate School of Medicine for their technical assistants for immunoblot and FACS analysis, the Teijin Scholarship Foundation, Suntory Foundation for Life Science and the Japan Society for the Promotion of Science (JSPS) DC2 for scholarships to H.W. Japan Society for the Promotion of Science (JSPS) #19H03040 (R.S.), #22H02430 (R.S.), #21K08376 (M.I.), #21K08405 (N.M.), #21K08424 (N.K.), #19K08848 (M.A.), #22H02252 (Y.T.), #22H02915 (Y.T.), #22K20589 (H.W.), and Grant-in-aid for JSPS fellows #20J11377(H.W.) for fundings. We also thank funding from Ikeda Scientific Co., Ltd (H.W.), the SENSHIN Medical Research Foundation (M.A.) and Takeda Science Foundation (M.A.). We are grateful to Chuuk State Department of Marine Resources for providing permission to collect sponges.

## Author contributions
This work was conceptualized by H.W., R.S., M.A., and Y.T. H.N., T.Os., T.M., Y.K., T.Og., and H.W. analyzed the protein sequence. H.K., K.O., PJ.F., T.Y., T.O., D.F., and Y.T. overexpressed the proteins and analyzed their crystal structures and biochemical properties. N.M., M.A., M.I., N.K., and H.W. characterized the biological activities. K.H. and M.H. isolated hypnin. R.S., H.W., Y.T., M.A., and N.M. prepared the manuscript with input from all of the other authors.

## Competing interests
Araki and Imai are employees of Meiji Seika Pharma, and Komatsu has received a salary from Pharmaessentia Japan where he is a board member. The remaining authors declare no other competing interests.
