## [Peer Review File · Nature Communications]

A marine sponge-derived lectin reveals hidden pathway for thrombopoietin receptor activationREVIEWER COMMENTS

Reviewer #1 (Remarks to the Author):

This paper provides biochemical, structural, and thermodynamic evidence of how a marine sponge-derived thrombocortin (ThC) activates the thrombopoietin receptor (MPL). The authors reported the three-dimensional structure of ThC as a novel fucose-binding lectin and the mechanisms underlying its MPL activation by binding to sugar chains on MPL. Moreover, this study showed the possibility that fucose-binding lectins could be used as novel tools to control structure to activate the MPL complex and enable fine-tuning of dimerization, internalization, and signaling in conjunction with coapplication with other agonists.

The experiments in this manuscript are well designed and the obtained results are properly presented and discussed. However, I think that the authors can discuss the mechanism of MPL activation more deeply by improving the analysis of the obtained ITC data. I recommend accepting the manuscript after comments described below are considered in the revision.

[Major comment (Page 18, line 384-388)]

"When the core-(1,6) fucose of the sugar chain at N117 of each MPL bound tightly to two ligand binding cores of the lectin, the configuration of the lectin-bound receptor complex directly reflected the spatial relationship of the pocket. Therefore, ThC- and PA-IIL-bound activating receptor complexes differ structurally. We propose that this structural difference affects the efficacies of the two lectins." This author's proposal is based on the difference in the relative positions of two fucose molecules bound to the ThC dimer and PA-IIL dimer in the crystal structures (Fig. 4C). On the other hand, ITC experiments (Table S1) exhibited that the thermodynamic mechanisms of the rThC/fucose binding reaction and the PA-IIL/fucose binding reaction were significantly different. Negative ΔH and positive ΔS of the PA-IIL/fucose binding reaction indicate favorable hydrogen bonding and hydrophobic interactions. Conversely, negative ΔH and negative ΔS of the rThC/fucose binding reaction indicate the enthalpy driven process and unfavorable ΔS , suggesting that some conformational change may occur during the binding reaction. Interestingly, in the case of the mannose binding reaction, these thermodynamic characteristics of ThC and PA-IIL are reversed (Table S1). Is there a possibility that the induced-fit conformational changes occur in the rThC/fucose-binding and the PA-IIL/mannose-binding reactions, but not in the rThC/mannose-binding and the PA-IIL/fucose-binding reactions? Is it possible that these differences in the mechanism of sugar binding assist the sugar-selective complexation and the suitable configuration of the lectin-bound receptor complex? More detailed inspections of the obtained thermodynamic and structural data could provide a deeper insight into the differences in the mechanisms of MPL activation by ThC and PA-IIL (and should be).

[Minor comment 1] (Page 8, line 141-143)

Is His-tag of rThC in a disordered state or could it be located near the fucose binding site? The state of the His-tag in the crystal structure should be described to express that the His-tag does not affect the fucose binding site. Similarly, it is better to describe whether or not the fucose binding site is affected by the neighboring ThC in the crystal packing.

[Minor comment 2] (Page 20, line 421-423)

"The strong synergy of ThC with TPO supports this hypothesis because predimerization facilitated by ThC largely reduces entropy during dimer formation."

The description after "because..." is conceptual but important to discuss the MPL activation mechanism synergistically caused by ThC and TPO. If possible, please cite any articles or data showing an example that dimerization or multimerization reduces entropy. (Examples of other proteins are also acceptable.)

[Minor comment 3] (Page 25, line 556-560)

The values of thermodynamic parameters measured by ITC vary with temperature and solvent conditions. Also, a Tris buffer, for example, is not suitable for determining thermodynamic parameters accurately by ITC, because the binding of proton to Tris itself causes a large enthalpy change

(Journal of Physical and Chemical Reference Data 31, 231 (2002); <https://doi.org/10.1063/1.1416902>). Therefore, it is necessary to describe temperature and solvent conditions in the “Methods”, Fig. 2D, Table S1, supplementary Fig. 8 and supplementary Fig. 11. ΔG values should also be added to Table S1.

[Minor comment 4] (Page 26, line 572-578)

Crystallization method (Hanging drop method, sitting drop method, etc.) and temperature condition for crystallization should be described.

[Minor comment 5] (Fig. 4)

The title of Fig.4 is “Involvement of the fucose moiety in ThC-dependent MPL activation.”. However, Fig. 4B shows the result of PA-IIL. Is it possible to add the result of ThC (Fig. 1B?) into Fig. 4B in order to compare the results of ThC and PA-IIL?

Reviewer #2 (Remarks to the Author):

The manuscript by Watari and co-workers highlights the potential functional mechanisms of a novel sea sponge-derived lectin (Thrombocortin; ThC) that appears to activate the thrombopoietin receptor, MPL. This manuscript adds to their previous work published in 2019 (PMID: 30978513) where they first identified that the purified form of ThC was able to drive proliferation and signalling in MPL-expressing cell lines. This more recent work provides a greater insight in the to potential mechanisms of action, identifying the importance of calcium-dependent fructose binding and ThC homodimerization leading the authors to suggest that ThC may actually activate MPL using a mechanism similar to that of mutant calreticulin; one of the primary driver mutations in MPNs. They also identify interesting differences in the signalling kinetics between ThC and TPO; with ThC providing a more delayed and somewhat more prolonged activation of JAK2, rather than an early brief peak of phosphorylation that is characteristic in TPO stimulation. They suggest that this may be down to reduced receptor internalization following stimulation.

Overall, I found this manuscript interesting and the results are generally robust. However, I feel that there are certain key mechanistic hypotheses that are eluded to without being backed up with experimental data.

- I would encourage the authors to characterise the extent by which ThC can stimulate MPL dimerization. This could be achieved using multiple different methods such as FRET, Nano-BiT, PLA or super resolution microscopy. Although the data points towards this being the mechanism of action, this has to be clearly demonstrated experimentally.
- To significantly advance the previous paper, in which the same Ba/F3 cell lines were stimulated with ThC, I would suggest determining the function in primary cells. It would be important and interesting to determine whether the differences in activation kinetics alter, for example, megakaryocyte differentiation or HSC expansion. This could also be further improved by determining any in vivo functionality in mouse models – comparing the effects with TPO.
- The signalling assays remain somewhat limited. The authors should consider expanding this to include activation of non-JAK/STAT pathways for determine any differences in other pathways. This would be interesting considering the differences in signalling between the most common TPO agonists eltrombopag and romiplostim.

Other, more minor improvements that could be made:

- Throughout the manuscript you use a lot more ThC compared with TPO (usually ~500-1000 fold). Why is this? The experiments where you compare activity/functionality would be better if you compared concentrations (e.g. in nM) than amounts.
- Fig. 4A – Why is there a significant reduction in the proliferation with 10 μ g/ml ThC compared to 1 μ g/ml? Is this expected?
- Fig. 4D – This is interesting and important data – but why switch to HEK cells? It’s clear that you

have a good Ba/F3-MPL model up and running, why not introduce the mutations in these cells instead? This would allow you to compare more signalling/growth with a direct comparison to your previous data and in a slightly more physiologically relevant cell line. It is likely that the receptor density, diffusion and signalling will be different in the HEK cells.

- Fig. 5A – Can the authors propose a reason why there seems to be a disconnect between pJAK2 and pSTAT5 in the kinetics of ThC activity? pSTAT5 seems to peak at 30mins, which pJAK2 doesn't peak until 360mins. Could this be something to do with the activation of negative regulators? If so, it would be good to identify any changes.
- Fig 5B – The internalization data is interesting, but I'm not sure why this method (lysis then membrane protein extraction) was used over flow cytometry. Would this not give more robust and consistent results? Also, the authors cite their previous publication in which this method was used (PMID: 31462733) – but in this paper they show the actual western blot data in addition to the graphs. This would significantly add to the interpretation of this data.
- As the authors propose that ThC uses similar mechanisms to activate MPL and mutCALR, it would be interesting to see if the presence of mutCALR alters the effects of ThC. Presumably there would be competition for binding to MPL?

Reviewer #3 (Remarks to the Author):

Hidden pathway for cytokine receptor activation: a marine sponge-derived lectin reveals sugar-mediated thrombopoietin receptor activation

This is a fascinating work that defines a lectin isolated from the marine sponge as a thrombopoietin synergist. The mechanism of action is related to retained thrombopoietin receptor expression and subsequent thrombopoietin binding.

The work is solid with very advanced protein chemical analysis, crystallization, dose response, and signal transduction. Moreover the work identifies an important link of receptor activation in the CALR mutant context by implicating the same N glycosylation site as activating.

This is an exciting study given the breadth of studies that define this unique lectin isolated from the marine sponge and its unusual effect on thrombopoietin signaling. In addition to the insights provided into CALR MPN mutant context, this also affords an opportunity for improved thrombopoietin delivery as a therapeutic.

Some limitations of this study:

- 1) Some elaboration as to the process by which the marine sponge was selected to study lectins. Was this part of a broad lectin analysis, or search for thrombopoietins? Were many other organisms part of a screen to identify lectins with thrombopoietic properties
- 2) The lectin isolated by the sponge – does this have any similarity of human F-type lectins? Is there a homologous lectin in humans that may have physiologic consequences?

REVIEWER COMMENTS (Original)

Reviewer #1 (Remarks to the Author):

This paper provides biochemical, structural, and thermodynamic evidence of how a marine sponge-derived thrombocortecin (ThC) activates the thrombopoietin receptor (MPL). The authors reported the three-dimensional structure of ThC as a novel fucose-binding lectin and the mechanisms underlying its MPL activation by binding to sugar chains on MPL. Moreover, this study showed the possibility that fucose-binding lectins could be used as novel tools to control structure to activate the MPL complex and enable fine-tuning of dimerization, internalization, and signaling in conjunction with coapplication with other agonists.

The experiments in this manuscript are well designed and the obtained results are properly presented and discussed. However, I think that the authors can discuss the mechanism of MPL activation more deeply by improving the analysis of the obtained ITC data. I recommend accepting the manuscript after comments described below are considered in the revision.

[Major comment (Page 18, line 384-388)]

"When the core-(1,6) fucose of the sugar chain at N117 of each MPL bound tightly to two ligand binding cores of the lectin, the configuration of the lectin-bound receptor complex directly reflected the spatial relationship of the pocket. Therefore, ThC- and PA-IIL-bound activating receptor complexes differ structurally. We propose that this structural difference affects the efficacies of the two lectins."

This author's proposal is based on the difference in the relative positions of two fucose molecules bound to the ThC dimer and PA-IIL dimer in the crystal structures (Fig. 4C). On the other hand, ITC experiments (Table S1) exhibited that the thermodynamic mechanisms of the rThC/fucose binding reaction and the PA-IIL/fucose binding reaction were significantly different. Negative ΔH and positive ΔS of the PA-IIL/fucose binding reaction indicate favorable hydrogen bonding and hydrophobic interactions. Conversely, negative ΔH and negative ΔS of the rThC/fucose binding reaction indicate the enthalpy driven process and unfavorable ΔS , suggesting that some conformational change may occur during the binding reaction. Interestingly, in the case of the mannose binding reaction, these thermodynamic characteristics of ThC and PA-IIL are reversed (Table S1). Is there a possibility that the induced-fit conformational changes occur in the rThC/fucose-binding and the PA-IIL/mannose-binding reactions, but not in the rThC/mannose-binding and the PA-IIL/fucose-binding reactions? Is it possible that these differences in the mechanism of sugar binding assist the sugar-selective complexation and the suitable configuration of the lectin-bound receptor complex? More detailed inspections of the obtained

thermodynamic and structural data could provide a deeper insight into the differences in the mechanisms of MPL activation by ThC and PA-IIL (and should be).

Thank you for your thoughtful comment. As the reviewer suggested, we have compared the crystal structures of apo-rThC, rThC-fucose, rThC-mannose, apo-PAIIL, PAIIL-fucose, and PAIIL-mannose. However, no major structural changes associated with fucose or mannose binding were observed (Table R1, Figure R1; see below).

Table R1. Conformational changes and entropy terms associated with lectins upon sugar binding.

		Fuc	Man
rThC	global ^a	0.21	0.17
	local ^b	large	large
	ΔS	-2.94	16.6
PA-IIL	global ^a	0.39	0.21
	local ^b	larger	slight
	ΔS	3.75	-12.9

- a. RMSD (Å), rThC: apo vs fucose, 218 C α atoms; apo vs mannose, 214 C α atoms
 PA-IIL (when comparing dimers): apo vs. fucose, 215 C α atoms; apo vs. mannose, 200 C α atoms.
 b. Based on the conformational difference between the apo- and sugar-bound states around the sugar-binding sites in Figure R1.

The RMSDs for the apo- and sugar-bound structures were very low in all cases, suggesting that there were no major global structural changes interpretable as induced fit, which is typically considered when the RMSD is greater than ~ 2 Å (Sherman *et al.* J. Med. Chem. 49.2 (2006): 534-553).

Some local structural changes, however, in both rThC and PA-IIL were observed, but only at residues around the sugar binding site upon fucose/mannose binding. However, no clear correlation between mannose and fucose and the structural changes was found (Table R1); that is, for rThC, both fucose/mannose showed conformational changes compared with the apo form, whereas for PA-IIL, mannose-bound forms showed only slight differences from the apo state, but the fucose-bound form appeared to undergo a conformational change. However, no clear correlation was found between conformational changes and entropy.

This result suggests that it is difficult to interpret each interaction solely based on the numerical thermodynamic properties obtained in each experiment. In fact, one study (Pokorná *et al. Biochemistry* 45.24 (2006): 7501-7510) investigated, in detail, the correlation between sugar-binding activity and the conformation of sugar-binding proteins (PA-IIL and CV-IIL), which are very similar to rThC. Sabin *et al.* also reported a slight gain in entropy in binding between PA-IIL and fucose, whereas for that with methyl- α -fucose, unfavorable entropy was evident (Sabin, *et al. FEBS letters* 580.3 (2006): 982-987). In both cases, no obvious correlation between the structure and thermodynamic profiles was found, and the authors of both papers concluded that it was difficult to corroborate the mechanistic basis at this point.

Moreover, the limitations of ITC experiments should also be considered. When the ligand–protein interaction is weak, the thermodynamic parameters obtained in the ITC experiment are not perfectly suited to discuss the detailed binding modes. For example, the thermogram for ThC/fucose gave an ideal sigmoid curve, whereas that of ThC/mannose was considerably poorer owing to weak binding (Suppl Fig S8), and the latter case caused an inaccurate measurement of thermodynamic parameters.

Finally, when the difference in agonist actions was considered, the thermodynamic parameter with monosaccharide alone could not account for the ‘real’ binding between complex sugar chains in MPL. To further discuss this issue, structural and thermodynamic data based on the association between lectin and MPL are required; however, these are beyond the scope of this manuscript.

Figure R1. Superimposition of rThC and PA-IIL ligand in unbound (orange), fucose-bound (red), and mannose-bound (blue) states. The upper panel shows the global structure, and the lower panel shows the local structure around the sugar binding site.

Considering all of the above issues together, we added the following descriptions to the main text and added ‘Supplemental Note 2’ in the Supplementary Information to discuss the above interesting (but elusive) thermodynamic and structural nature of these lectins.

P12, line 222

“We thus compared the crystal structures of rThC and PA-IIL with or without sugars since some differences in thermodynamic profiles between the two lectins in ITC experiments suggested discrete modes of sugar binding (Table S1). However, no obvious differences that could result in different agonist actions were found (Supplementary Note 2).”

[Minor comment 1] (Page 8, line 141-143)

Is His-tag of rThC in a disordered state or could it be located near the fucose binding site? The state of the His-tag in the crystal structure should be described to express that the His-tag does not affect the fucose binding site. Similarly, it is better to describe whether or not the fucose binding site is affected by the neighboring ThC in the crystal packing.

Thank you for your comments. The His-tag fused to the N-terminus was disordered and could not be mapped to the crystal structure. As mentioned in the text (P10), the carboxyl group at the C-terminus was determined to be the most important region for sugar binding; however, the N-terminus was found to be in a completely different position from the sugar-binding site and is not expected to affect sugar binding.

Figure R2. Stereo ribbon diagram of the ThC dimer in a complex with fucose. The N-terminus, C-terminus, and bound fucose are shown as arrows. One protomer is colored according to the sequence based on a rainbow color ramp going from blue at the N-terminus to red at the C-terminus. Bound fucose and Ca ions are also shown as a ball-and-stick model.

Therefore, in this study, ThC with a His-tag fused to the N-terminus was used as rThC. It is evident that the addition of the His-Tag did not interfere with ThC binding to the receptor, because it showed agonist activity that was equivalent to that of the natural ThC. In the revised manuscript, the following description of His-Tag is provided on p. 25 of the Methods section.

P26, line 566

“The 6× His tag was attached to the N-terminus of ThC because the crystal structure demonstrated that the N-terminus of ThC and BC2LC is located far from the sugar-binding site. Therefore, the His-tag attached to the N-terminus does not affect fucose binding.”

Similarly, it is better to describe whether or not the fucose binding site is affected by the neighboring ThC in the crystal packing.

As the reviewer pointed out, some observations suggested that crystal packing affected the binding site. In the mannose–ThC complex, the sugar was bound to only one of the two carbohydrate-binding sites in the dimer. For ThC–fucose, the electron density at one of the carbohydrate binding

sites was obscurer than the other (average of B-factor of all atoms in the fucose was 8.4 and 22.6, respectively.) (Figure for Note 3). From these results, we postulate that under crystallization conditions, the two binding sites in these lectins are not perfectly equivalent due to crystal packing. However, 1:1 binding between ThC and fucose was clearly observed in the ITC experiment, where the thermogram showed an ideal two-state sigmoidal curve. Thus, it is clear that the two sugar-binding sites are equivalent in solution. Given that fucose/mannose was introduced to the sugar-binding site via co-crystallization, not through soaking into the rThC crystal, it is reasonable to consider that the crystallization condition is preferable for fucose/mannose binding to only one sugar-binding site; that is, in the crystallization solution, crystal growth and packing occurred when the sugar was bound to one site, rather than both binding sites. In the revised manuscript, we have added a description of this uneven binding to Supplementary Note 3, which is linked to the main text (P9, line 170).

Supplementary Note 3.

“In the crystal structure of rThC–fucose, the electron density of fucose at one of the carbohydrate-binding sites was more obscure than that at the other (the average B-factor of all atoms in fucose was 8.4 and 22.6, respectively; Figure for Note 3). Moreover, mannose clearly bound to one sugar-binding site but not to the other (data not shown). These results suggest that the binding of sugars to the two sites in the crystal milieu is not equivalent, likely owing to neighboring effects in the process of crystal packing. However, the ITC thermogram, showing a sigmoidal curve with a typical two-state transition, indicated that the stoichiometry of ThC to sugar was 1:1, and two sugar-binding sites in the ThC dimer were occupied in solution. Therefore, the structure of the site with clear electron density was used in this study to investigate the interaction between sugar and ThC.

[Minor comment 2] (Page 20, line 421-423)

“The strong synergy of ThC with TPO supports this hypothesis because predimerization facilitated by ThC largely reduces entropy during dimer formation.”

The description after "because..." is conceptual but important to discuss the MPL activation mechanism synergistically caused by ThC and TPO. If possible, please cite any articles or data showing an example that dimerization or multimerization reduces entropy. (Examples of other proteins are also acceptable.)

Thank you for your comments. We found examples of both entropically favorable (Philo *et al.*

Biochemistry 35.5 (1996): 1681-1691) and unfavorable (Verstraete *et al* *Blood*, 118.1 (2011): 60-68) dimer formation reported in the literature based on other hematopoietic receptors. Thus, the thermodynamic course depends on the specific dimerization mechanism, and the actual thermodynamic terms cannot be predicted without experimental data. We thus instead suggested here that a non-covalent ternary complex formed between the lectin and two receptor monomers ('predimerized form, Fig 5D') would help to largely increase the probability of MPL being activated immediately upon TPO binding (as opposed to TPO binding to a monomeric receptor) by avoiding the cost to meet with the other half of the receptor. We have rewritten this sentence as follows to convey our idea more comprehensively.

Main text P21, lines 445-447.

Before (Blue part)

The role of this well-conceivable transition complex formed in the middle of the activation process was previously hidden because this intermediate is short-lived in TPO-activated MPL due to potent 'native' interactions between the receptor and ligand²⁶. The strong synergy of ThC with TPO supports this hypothesis because predimerization facilitated by ThC largely reduces entropy during dimer formation. Therefore, ThC stabilizes transition state III, although ThC-bound III eventually shifts to form signaling complex IV, presumably via the aid of intrinsic domain interactions in the transmembrane (TMD) and intracellular domains (Fig. 5D).

After revision (Yellow part)

"The role of this well-conceivable transition complex formed in the middle of the activation process was previously hidden because this intermediate is short-lived in TPO-activated MPL due to potent 'native' interactions between the receptor and ligand²⁶. The strong synergy of ThC with TPO supports the presence of this transition state because predimerization reduces activation barrier existing in between the state II and III (Fig. 5D). Therefore, ThC stabilizes transition state III, although ThC-bound III eventually shifts to form signaling complex IV, presumably via the aid of intrinsic domain interactions in the transmembrane (TMD) and intracellular domains (Fig. 5D).

[Minor comment 3] (Page 25, line 556-560)

The values of thermodynamic parameters measured by ITC vary with temperature and solvent conditions. Also, a Tris buffer, for example, is not suitable for determining thermodynamic parameters accurately by ITC, because the binding of proton to Tris itself causes a large enthalpy change (*Journal of Physical and Chemical Reference Data* 31, 231 (2002); <https://doi.org/10.1063/1.1416902>). Therefore, it is necessary to describe temperature and solvent conditions in the "Methods", Fig. 2D, Table S1, supplementary Fig. 8 and supplementary

Fig. 11. ΔG values should also be added to Table S1.

Thank you for your suggestion. In our experiments, 20 mM HEPES-Na (pH 8.0) and 200 mM NaCl were used for ITC instead of Tris buffer at 25°C. Information on temperature and buffer conditions for ITC measurements has now been provided in the Methods (p27, line 591), Fig. 2D (legend), Table S1, Supplementary Fig. 8, and Supplementary Fig. 11. The ΔG values are listed in Table S1.

[Minor comment 4] (Page 26, line 572-578)

Crystallization method (Hanging drop method, sitting drop method, etc.) and temperature condition for crystallization should be described.

Thank you for your suggestion. We have added the following description of crystallization conditions to Methods.

“Crystallization was carried out by sitting-drop vapor diffusion method at 20 °C”

[Minor comment 5] (Fig. 4)

The title of Fig.4 is “Involvement of the fucose moiety in ThC-dependent MPL activation.”. However, Fig. 4B shows the result of PA-IIL. Is it possible to add the result of ThC (Fig. 1B?) into Fig. 4B in order to compare the results of ThC and PA-IIL?

Thank you for your suggestion. We added the overlay of ThC data in Fig. 4B.

Reviewer #2 (Remarks to the Author):

The manuscript by Watari and co-workers highlights the potential functional mechanisms of a novel sea sponge-derived lectin (Thrombocortin; ThC) that appears to activate the thrombopoietin receptor, MPL. This manuscript adds to their previous work published in 2019 (PMID: 30978513) where they first identified that the purified form of ThC was able to drive proliferation and signalling in MPL-expressing cell lines. This more recent work provides a greater insight in the to potential mechanisms of action, identifying the importance of calcium-dependent fructose binding and ThC homodimerization leading the authors to suggest that ThC may actually activate MPL using a

mechanism similar to that of mutant calreticulin; one of the primary driver mutations in MPNs. They also identify interesting differences in the signalling kinetics between ThC and TPO; with ThC providing a more delayed and somewhat more prolonged activation of JAK2, rather than an early brief peak of phosphorylation that is characteristic in TPO stimulation. They suggest that this may be down to reduced receptor internalization following stimulation.

Overall, I found this manuscript interesting and the results are generally robust. However, I feel that there are certain key mechanistic hypotheses that are eluded to without being backed up with experimental data.

- I would encourage the authors to characterise the extent by which ThC can stimulate MPL dimerization. This could be achieved using multiple different methods such as FRET, Nano-BiT, PLA or super resolution microscopy. Although the data points towards this being the mechanism of action, this has to be clearly demonstrated experimentally.

Thank you for bringing this point to our attention. As mentioned in the Discussion, P21 lines 440–447, we believe that the role of ThC is to stabilize the dimeric state by crosslinking the sugar chain of two monomeric receptors. As demonstrated by Wilmes et al. (ref 26), MPL exists in a monomeric form, but upon ligand binding, it forms a ligand-bound dimeric activation complex. In this process, the probability of dimerization (including the formation of an inactivated dimer) is independent of the ligand but is proportional to the receptor population. We propose that ThC stabilizes the dimeric state of the receptor through the formation of a ternary complex. We would love to try single-molecule imaging microscopy in conjunction with fluorescent techniques to confirm our hypothesis, as you suggested. However, I think a new set of experiments would be required for this, and thus, it is beyond the scope of the present manuscript. I have sought opportunities to conduct such studies using truly unique MPL ligands, including ThC.

- To significantly advance the previous paper, in which the same Ba/F3 cell lines were stimulated with ThC, I would suggest determining the function in primary cells. It would be important and interesting to determine whether the differences in activation kinetics alter, for example, megakaryocyte differentiation or HSC expansion. This could also be further improved by determining any in vivo functionality in mouse models – comparing the effects with TPO.

We agree with the reviewer's suggestion. Therefore, we performed an in vitro assay with induced pluripotent stem cells and examined the potential of ThC to promote megakaryocytic differentiation. However, the assay was perturbed by cell aggregation induced by ThC, which harbors agglutinating activity. We analyzed the aggregated cells, but

no obvious megakaryocyte differentiation was observed in the presence of ThC (data not shown). Because this is an important issue, we have considered it in the Discussion section.

P18, line 381

“To validate the potential of ThC as an agonist for MPL, we performed an in vitro assay with human hematopoietic stem cells derived from induced pluripotent stem cells and examined the potential of ThC to induce megakaryocytic differentiation. However, the assay was perturbed by cell aggregation induced by ThC, which harbors agglutinating activity. We analyzed the aggregated cells, but no obvious megakaryocyte differentiation was observed in the presence of ThC (data not shown).”

- The signalling assays remain somewhat limited. The authors should consider expanding this to include activation of non-JAK/STAT pathways for determine any differences in other pathways. This would be interesting considering the differences in signalling between the most common TPO agonists eltrombopag and romiplostim.

Based on your suggestion, we examined the phosphorylation status of proteins of pathways other than STAT. As a result, the levels of activated pERK and pAKT seemed to be lower than those of pSTAT5 in ThC-treated cells, suggesting a potential difference in downstream activation. This difference could be caused by the different mode of MPL activation by ThC, which did not couple with internalization of the receptor. Interestingly, eltrombopag resulted in more robust activation of STAT5 than TPO, suggesting a more diverse mechanism of MPL downstream activation dependent on agonists. We have added these data to Supplementary Fig. S1d of the revised manuscript.

Other, more minor improvements that could be made:

- Throughout the manuscript you use a lot more ThC compared with TPO (usually ~500-1000 fold). Why is this? The experiments where you compare activity/functionality would be better if you compared concentrations (e.g. in nM) than amounts.

This is due to the difference in intrinsic potency between TPO ($EC_{50} = 2.8 \text{ ng/mL}$, 150 pM, P16, line 339) and rThC ($EC_{50} = 0.26 \text{ } \mu\text{g/mL}$, 18.6 nM, P6, line 9, figure legend). We now included the information of concentration in the manuscript.

- Fig. 4A – Why is there a significant reduction in the proliferation with 10µg/ml ThC compared to 1µg/ml? Is this expected?

Thank you for your comment. We indeed expected this result.

The suppression of thrombopoietin receptors via an excess amount of agonist has been observed in previous studies (Yoshida et al. *Exp. Hematol.* 59 (2018): 30-39., Xie *et al. J. Cell. Mol. Med.* 22.11 (2018): 5367-5377) and reviewed by Hitchcock (*Platelets* 32, 770-778 (2021), Ref 4 of this manuscript). This is likely due to saturation of the ligand-binding domain of a receptor molecule on the cell surface by a ligand, which interferes with ligand binding to another receptor, perturbing receptor dimerization and activation. In addition, ThC has agglutinating activity at high concentrations, which might have a negative effect on cell proliferation. To explain this, we have added the following line to the legend of Figure 4A.

“Note that cell proliferation was suppressed in the presence of 10 µg/mL ThC, presumably due to the suppression of cell proliferation via its agglutinating activity and due to interference with receptor dimerization mediated by the occupation of a ligand-binding site on a receptor molecule.”

- Fig. 4D – This is interesting and important data – but why switch to HEK cells? It's clear that you have a good Ba/F3-MPL model up and running, why not introduce the mutations in these cells instead? This would allow you to compare more signalling/growth with a direct comparison to your previous data and in a slightly more physiologically relevant cell line. It is likely that the receptor density, diffusion and signalling will be different in the HEK cells.

We used a reporter assay with HEK cells to investigate N-glycan involvement because this transfection-based assay circumvented the need to establish a cell line and allowed us to examine significantly more diverse mutant forms of MPL in a short time. We agree with your concern regarding the use of different systems to investigate ThC-dependent MPL activation. However, the HEK reporter system has been used to examine MPL ligands, including mutCALR (Shide, *et al. Leukemia* (2017): 1136-1144. Araki *et al. Leukemia* 33.1 (2019): 122-131, Levy *et al. Blood* 135.12 (2020): 948-953), and the consistency between the transient (HEK and other cell lines) and stable (Ba/F3 and other cell lines) expression systems has been recognized. We understand your concern; thus, we describe the reasons for using the HEK system in the Methods section.

“To determine the critical site for ThC activation, mutants in which the N residues were replaced by Q residues were expressed in HEK293T cells, and receptor activation was monitored using the STAT5 reporter assay, which has been recognized as a model system to validate MPL ligands^{21, 36, 37}”

- Fig. 5A – Can the authors propose a reason why there seems to be a disconnect between pJAK2 and pSTAT5 in the kinetics of ThC activity? pSTAT5 seems to peak at 30mins, which pJAK2 doesn't peak until 360mins. Could this be something to do with the activation of negative regulators? If so, it would be good to identify any changes.

We believe that the different kinetics of pJAK2 and pSTAT5 between TPO and ThC might reflect the availability of phosphatase around JAK2. For TPO, the receptor complex with JAK2 is rapidly internalized, which might induce destruction of the complex and the dephosphorylation of JAK2. For ThC, the receptor complex remained on the cell surface, where the phosphatase might be less abundant and/or suppressed because of the formation of a complex comprising JAK2 with the receptor. In contrast, pSTAT5 is rapidly dephosphorylated, as its regulation is different from the localization of the receptor and JAK2 complex. Furthermore, in the presence of ThC, JAK2 remained on the cell surface along with MPL, and levels of autophosphorylation were slowly increased. Alternatively, the slow and steady activation of JAK2 without the internalization of MPL mediated by ThC might not be associated with the strong phosphorylation of canonical Y1007/1008 residues. Nevertheless, a more detailed analysis is required to elucidate the molecular mechanism underlying JAK2 and downstream activation mediated by lectin ligands.

- Fig 5B – The internalization data is interesting, but I'm not sure why this method (lysis then membrane protein extraction) was used over flow cytometry. Would this not give more robust and consistent results? Also, the authors cite their previous publication in which this method was used (PMID: 31462733) – but in this paper they show the actual western blot data in addition to the graphs. This would significantly add to the interpretation of this data.

Only a few antibodies are suitable for the flow cytometric analysis of MPL. Furthermore, our pilot experiments suggested potential interference with antibody binding to MPL mediated by the ligand. Therefore, we used a biochemical approach to quantify surface receptor levels. With respect to the western blot data, we have now provided these in supplemental Fig. 14.

- As the authors propose that ThC uses similar mechanisms to activate MPL and mutCALR, it

would be interesting to see if the presence of mutCALR alters the effects of ThC. Presumably there would be competition for binding to MPL?

This is an interesting question. However, to date, we (and perhaps others) have failed to perform a binding assay for mutCALR and MPL with purified proteins. Results of co-immunoprecipitation assay using cell lysate with these proteins co-expressed showed that mutCALR interacts with MPL harboring the immature form of N-glycan, which is predominantly present in the endoplasmic reticulum (ER) (Chachoua, *et al.* *Blood*. 2016;127(10):1325-35, Masubuchi, *et al.* *Leukemia*. 2020;34(2):499-509). These observations suggest that the interaction between mutCALR and MPL requires conditions only present in the ER, which need to be defined to perform the competition assay.

Reviewer #3 (Remarks to the Author):

Hidden pathway for cytokine receptor activation: a marine sponge-derived lectin reveals sugar-mediated thrombopoietin receptor activation

This is a fascinating work that defines a lectin isolated from the marine sponge as a thrombopoietin synergist. The mechanism of action is related to retained thrombopoietin receptor expression and subsequent thrombopoietin binding.

The work is solid with very advanced protein chemical analysis, crystallization, dose response, and signal transduction. Moreover, the work identifies an important link of receptor activation in the CALR mutant context by implicating the same N glycosylation site as activating.

This is an exciting study given the breadth of studies that define this unique lectin isolated from the marine sponge and its unusual effect on thrombopoietin signaling. In addition to the insights provided into CALR MPN mutant context, this also affords an opportunity for improved thrombopoietin delivery as a therapeutic.

Some limitations of this study:

1) Some elaboration as to the process by which the marine sponge was selected to study lectins. Was this part of a broad lectin analysis, or search for thrombopoietins? Were many other organisms part of a screen to identify lectins with thrombopoietic properties

Thank you for your comments. In our previous publication (Ref 12 of the manuscript), we aimed

to identify thrombopoietic compounds and screened approximately 900 marine organisms. ThC was found to be an active molecule and one of the most potent extracts. However, the hit rate was low, and only four other active specimens were identified.

2) The lectin isolated by the sponge – does this have any similarity of human F-type lectins? Is there a homologous lectin in humans that may have physiologic consequences?

F-type lectins are ~140 aa lectins with a jelly roll motif forming a trimer. Two trimers are stacked into hexamers to form functional lectins (Vasta *et al. Frontiers in Immunol.* 8 (2017): 1648). This structural motif is very different from that of ThC, which we do not expect to have biological activity like ThC. However, it is interesting to hypothesize that some human lectins act like ThC under normal physiological conditions. It could be a fucose-binding lectin, but lectins with other sugar selectivities might function in mechanisms such as those of CALRmut. The physiological importance of sugar-mediated MPL activation is an intriguing

=====

Point-by-point Responses to the Reviewers Comments

Manuscript ID: NCOMMS-22-10090A

Response to Reviewer #1

1. This author's proposal is based on the difference in the relative positions of two fucose molecules bound to the ThC dimer and PA-IIL dimer in the crystal structures (Fig. 4C). On the other hand, ITC experiments (Table S1) exhibited that the thermodynamic mechanisms of the rThC/fucose binding reaction and the PA-IIL/fucose binding reaction were significantly different. Negative ΔH and positive ΔS of the PA-IIL/fucose binding reaction indicate favorable hydrogen bonding and hydrophobic interactions. Conversely, negative ΔH and negative ΔS of the rThC/fucose binding reaction indicate the enthalpy driven process and unfavorable ΔS , suggesting that some conformational change may occur during the binding reaction. Interestingly, in the case of the mannose binding reaction, these thermodynamic characteristics of ThC and PA-IIL are reversed (Table S1). Is there a possibility that the induced-fit conformational changes occur in the rThC/fucose-binding and the PA-IIL/mannose-binding reactions, but not in the rThC/mannose-binding and the PA-IIL/fucose-binding reactions? Is it possible that these differences in the mechanism of sugar binding assist the sugar-selective complexation and the suitable configuration of the lectin-bound receptor complex? More detailed inspections of the obtained thermodynamic and structural data could provide a deeper insight into the differences in the mechanisms of MPL activation by ThC and PA-IIL (and should be).

Thank you for your thoughtful comment. As the reviewer suggested, we have compared the crystal structures of apo-rThC, rThC-fucose, rThC-mannose, apo-PAIIL, PAIIL-fucose, and PAIIL-mannose. However, no major structural changes associated with fucose or mannose binding were observed (Table R1, Figure R1; see below).

Table R1. Conformational changes and entropy terms associated with lectins upon sugar binding.

		Fuc	Man
rThC	global ^a	0.21	0.17
	local ^b	large	large
	ΔS	-2.94	16.6
PA-IIL	global ^a	0.39	0.21
	local ^b	larger	slight
	ΔS	3.75	-12.9

- c. RMSD (\AA), rThC: apo vs fucose, 218 C α atoms; apo vs mannose, 214 C α atoms
 PA-IIL (when comparing dimers): apo vs. fucose, 215 C α atoms; apo vs. mannose, 200 C α atoms.
 d. Based on the conformational difference between the apo- and sugar-bound states around the sugar-binding sites in Figure R1.

The RMSDs for the apo- and sugar-bound structures were very low in all cases, suggesting that there were no major global structural changes interpretable as induced fit, which is typically considered when the RMSD is greater than $\sim 2 \text{\AA}$ (Sherman *et al.* J. Med. Chem. 49.2 (2006): 534-553).

Some local structural changes, however, in both rThC and PA-IIL were observed, but only at residues around the sugar binding site upon fucose/mannose binding. However, no clear correlation between mannose and fucose and the structural changes was found (Table R1); that is, for rThC, both fucose/mannose showed conformational changes compared with the apo form, whereas for PA-IIL, mannose-bound forms showed only slight differences from the apo state, but the fucose-bound form appeared to undergo a conformational change. However, no clear correlation was found between conformational changes and entropy.

This result suggests that it is difficult to interpret each interaction solely based on the numerical thermodynamic properties obtained in each experiment. In fact, one study (Pokorná *et al.* *Biochemistry* 45.24 (2006): 7501-7510) investigated, in detail, the correlation between sugar-binding activity and the conformation of sugar-binding proteins (PA-IIL and CV-IIL), which are very similar to rThC. Sabin *et al.* also reported a slight gain in entropy in binding between PA-IIL and fucose, whereas for that with methyl- α -fucose, unfavorable entropy was evident (Sabin, *et al.* *FEBS letters* 580.3 (2006): 982-987). In both cases, no obvious correlation between the structure

and thermodynamic profiles was found, and the authors of both papers concluded that it was difficult to corroborate the mechanistic basis at this point.

Moreover, the limitations of ITC experiments should also be considered. When the ligand–protein interaction is weak, the thermodynamic parameters obtained in the ITC experiment are not perfectly suited to discuss the detailed binding modes. For example, the thermogram for ThC/fucose gave an ideal sigmoid curve, whereas that of ThC/mannose was considerably poorer owing to weak binding (Suppl Fig S8), and the latter case caused an inaccurate measurement of thermodynamic parameters.

Finally, when the difference in agonist actions was considered, the thermodynamic parameter with monosaccharide alone could not account for the ‘real’ binding between complex sugar chains in MPL. To further discuss this issue, structural and thermodynamic data based on the association between lectin and MPL are required; however, these are beyond the scope of this manuscript.

Figure R1. Superimposition of rThC and PA-IIL ligand in unbound (orange), fucose-bound (red), and mannose-bound (blue) states. The upper panel shows the global structure, and the lower panel shows the local structure around the sugar binding site.

Considering all of the above issues together, we added the following descriptions to the main text

and added ‘Supplemental Note 2’ in the Supplementary Information to discuss the above interesting (but elusive) thermodynamic and structural nature of these lectins.

P12, line 222

“We thus compared the crystal structures of rThC and PA-III with or without sugars since some differences in thermodynamic profiles between the two lectins in ITC experiments suggested discrete modes of sugar binding (Table S1). However, no obvious differences that could result in different agonist actions were found (Supplementary Note 2).”

2. Is His-tag of rThC in a disordered state or could it be located near the fucose binding site? The state of the His-tag in the crystal structure should be described to express that the His-tag does not affect the fucose binding site.

Thank you for your comments. The His-tag fused to the N-terminus was disordered and could not be mapped to the crystal structure. As mentioned in the text (P10), the carboxyl group at the C-terminus was determined to be the most important region for sugar binding; however, the N-terminus was found to be in a completely different position from the sugar-binding site and is not expected to affect sugar binding.

Figure R2. Stereo ribbon diagram of the ThC dimer in a complex with fucose. The N-terminus, C-terminus, and bound fucose are shown as arrows. One protomer is colored

according to the sequence based on a rainbow color ramp going from blue at the N-terminus to red at the C-terminus. Bound fucose and Ca ions are also shown as a ball-and-stick model.

Therefore, in this study, ThC with a His-tag fused to the N-terminus was used as rThC. It is evident that the addition of the His-Tag did not interfere with ThC binding to the receptor, because it showed agonist activity that was equivalent to that of the natural ThC. In the revised manuscript, the following description of His-Tag is provided on p. 25 of the Methods section.

P26, line 566

“The 6× His tag was attached to the N-terminus of ThC because the crystal structure demonstrated that the N-terminus of ThC and BC2LC is located far from the sugar-binding site. Therefore, the His-tag attached to the N-terminus does not affect fucose binding.”

3. Similarly, it is better to describe whether or not the fucose binding site is affected by the neighboring ThC in the crystal packing.

As the reviewer pointed out, some observations suggested that crystal packing affected the binding site. In the mannose–ThC complex, the sugar was bound to only one of the two carbohydrate-binding sites in the dimer. For ThC–fucose, the electron density at one of the carbohydrate binding sites was obscurer than the other (average of B-factor of all atoms in the fucose was 8.4 and 22.6, respectively.) (Figure for Note 3). From these results, we postulate that under crystallization conditions, the two binding sites in these lectins are not perfectly equivalent due to crystal packing. However, 1:1 binding between ThC and fucose was clearly observed in the ITC experiment, where the thermogram showed an ideal two-state sigmoidal curve. Thus, it is clear that the two sugar-binding sites are equivalent in solution. Given that fucose/mannose was introduced to the sugar-binding site via co-crystallization, not through soaking into the rThC crystal, it is reasonable to consider that the crystallization condition is preferable for fucose/mannose binding to only one sugar-binding site; that is, in the crystallization solution, crystal growth and packing occurred when the sugar was bound to one site, rather than both binding sites. In the revised manuscript, we have added a description of this uneven binding to Supplementary Note 3, which is linked to the main text (P9, line 170).

Supplementary Note 3.

“In the crystal structure of rThC–fucose, the electron density of fucose at one of the carbohydrate-binding sites was more obscure than that at the other (the average B-factor of all atoms in fucose was 8.4 and 22.6, respectively; Figure for Note 3). Moreover, mannose clearly bound to one sugar-binding site but not to the other (data not shown). These results suggest that the binding of sugars to the two sites in the crystal milieu is not equivalent, likely owing to neighboring effects in the process of crystal packing. However, the ITC thermogram, showing a sigmoidal curve with a typical two-state transition, indicated that the stoichiometry of ThC to sugar was 1:1, and two sugar-binding sites in the ThC dimer were occupied in solution. Therefore, the structure of the site with clear electron density was used in this study to investigate the interaction between sugar and ThC.

4. The strong synergy of ThC with TPO supports this hypothesis because predimerization facilitated by ThC largely reduces entropy during dimer formation.” The description after "because..." is conceptual but important to discuss the MPL activation mechanism synergistically caused by ThC and TPO. If possible, please cite any articles or data showing an example that dimerization or multimerization reduces entropy. (Examples of other proteins are also acceptable.)

Thank you for your comments. We found examples of both entropically favorable (Philo *et al. Biochemistry* 35.5 (1996): 1681-1691) and unfavorable (Verstraete *et al Blood*, 118.1 (2011): 60-68) dimer formation reported in the literature based on other hematopoietic receptors. Thus, the thermodynamic course depends on the specific dimerization mechanism, and the actual thermodynamic terms cannot be predicted without experimental data. We thus instead suggested here that a non-covalent ternary complex formed between the lectin and two receptor monomers (‘predimerized form, Fig 5D’) would help to largely increase the probability of MPL being activated immediately upon TPO binding (as opposed to TPO binding to a monomeric receptor) by avoiding the cost to meet with the other half of the receptor. We have rewritten this sentence as follows to convey our idea more comprehensively.

Main text P21, lines 445-447.

Before (Blue part)

The role of this well-conceivable transition complex formed in the middle of the activation process was previously hidden because this intermediate is short-lived in TPO-activated MPL due to potent ‘native’ interactions between the receptor and ligand²⁶. The strong synergy of ThC with TPO supports this hypothesis because predimerization facilitated by ThC largely reduces entropy during dimer formation. Therefore, ThC stabilizes transition state III, although ThC-bound III eventually shifts to form signaling complex IV, presumably via the aid of intrinsic domain interactions in the transmembrane (TMD) and intracellular domains (Fig. 5D).

After revision (Yellow part)

“The role of this well-conceivable transition complex formed in the middle of the activation process was previously hidden because this intermediate is short-lived in TPO-activated MPL due to potent ‘native’ interactions between the receptor and ligand²⁶. The strong synergy of ThC with TPO supports the presence of this transition state because predimerization reduces activation barrier existing in between the state II and III (Fig. 5D). Therefore, ThC stabilizes transition state III, although ThC-bound III eventually shifts to form signaling complex IV, presumably via the aid of intrinsic domain interactions in the transmembrane (TMD) and intracellular domains (Fig. 5D).

5. The values of thermodynamic parameters measured by ITC vary with temperature and solvent conditions. Also, a Tris buffer, for example, is not suitable for determining thermodynamic parameters accurately by ITC, because the binding of proton to Tris itself causes a large enthalpy change (Journal of Physical and Chemical Reference Data 31, 231 (2002); <https://doi.org/10.1063/1.1416902>). Therefore, it is necessary to describe temperature and solvent conditions in the “Methods”, Fig. 2D, Table S1, supplementary Fig. 8 and supplementary Fig. 11. ΔG values should also be added to Table S1.

Thank you for your suggestion. In our experiments, 20 mM HEPES-Na (pH 8.0) and 200 mM NaCl were used for ITC instead of Tris buffer at 25°C. Information on temperature and buffer conditions for ITC measurements has now been provided in the Methods (p27, line 591), Fig. 2D (legend), Table S1, Supplementary Fig. 8, and Supplementary Fig. 11. The ΔG values are listed in Table S1.

6. Crystallization method (Hanging drop method, sitting drop method, etc.) and temperature condition for crystallization should be described.

Thank you for your suggestion. We have added the following description of crystallization conditions to Methods.

"Crystallization was carried out by sitting-drop vapor diffusion method at 20 °C"

7. The title of Fig.4 is "Involvement of the fucose moiety in ThC-dependent MPL activation.". However, Fig. 4B shows the result of PA-III. Is it possible to add the result of ThC (Fig. 1B?) into Fig. 4B in order to compare the results of ThC and PA-III?

Thank you for your suggestion. We added the overlay of ThC data in Fig. 4B.

Response to Reviewer 2

8. I would encourage the authors to characterize the extent by which ThC can stimulate MPL dimerization. This could be achieved using multiple different methods such as FRET, Nano-BiT, PLA or super resolution microscopy. Although the data points towards this being the mechanism of action, this has to be clearly demonstrated experimentally.

Thank you for bringing this point to our attention. As mentioned in the Discussion, P21 lines 440–447, we believe that the role of ThC is to stabilize the dimeric state by crosslinking the sugar chain of two monomeric receptors. As demonstrated by Wilmes et al. (ref 26), MPL exists in a monomeric form, but upon ligand binding, it forms a ligand-bound dimeric activation complex. In this process, the probability of dimerization (including the formation of an inactivated dimer) is independent of the ligand but is proportional to the receptor population. We propose that ThC stabilizes the dimeric state of the receptor through the formation of a ternary complex. We would love to try single-molecule imaging microscopy in conjunction with fluorescent techniques to confirm our hypothesis, as you suggested. However, I think a new set of experiments would be

required for this, and thus, it is beyond the scope of the present manuscript. I have sought opportunities to conduct such studies using truly unique MPL ligands, including ThC.

9. To significantly advance the previous paper, in which the same Ba/F3 cell lines were stimulated with ThC, I would suggest determining the function in primary cells. It would be important and interesting to determine whether the differences in activation kinetics alter, for example, megakaryocyte differentiation or HSC expansion. This could also be further improved by determining any *in vivo* functionality in mouse models – comparing the effects with TPO.

We agree with the reviewer's suggestion. Therefore, we performed an *in vitro* assay with induced pluripotent stem cells and examined the potential of ThC to promote megakaryocytic differentiation. However, the assay was perturbed by cell aggregation induced by ThC, which harbors agglutinating activity. We analyzed the aggregated cells, but no obvious megakaryocyte differentiation was observed in the presence of ThC (data not shown). Because this is an important issue, we have considered it in the Discussion section.

P18, line 381

“To validate the potential of ThC as an agonist for MPL, we performed an *in vitro* assay with human hematopoietic stem cells derived from induced pluripotent stem cells and examined the potential of ThC to induce megakaryocytic differentiation. However, the assay was perturbed by cell aggregation induced by ThC, which harbors agglutinating activity. We analyzed the aggregated cells, but no obvious megakaryocyte differentiation was observed in the presence of ThC (data not shown).”

10. The signalling assays remain somewhat limited. The authors should consider expanding this to include activation of non-JAK/STAT pathways for determine any differences in other pathways. This would be interesting considering the differences in signalling between the most common TPO agonists eltrombopag and romiplostim.

Based on your suggestion, we examined the phosphorylation status of proteins of pathways other than STAT. As a result, the levels of activated pERK and pAKT seemed to be lower than those of pSTAT5 in ThC-treated cells, suggesting a potential difference in downstream activation. This difference could be caused by the different mode of MPL activation by ThC, which did not couple with internalization of the receptor. Interestingly, eltrombopag resulted

in more robust activation of STAT5 than TPO, suggesting a more diverse mechanism of MPL downstream activation dependent on agonists. We have added these data to Supplementary Fig. S1d of the revised manuscript.

11. Throughout the manuscript you use a lot more ThC compared with TPO (usually ~500-1000 fold). Why is this? The experiments where you compare activity/functionality would be better if you compared concentrations (e.g. in nM) than amounts.

This is due to the difference in intrinsic potency between TPO ($EC_{50} = 2.8 \text{ ng/mL}$, 150 pM, P16, line 339) and rThC ($EC_{50} = 0.26 \text{ } \mu\text{g/mL}$, 18.6 nM, P6, line 9, figure legend). We now included the information of concentration in the manuscript.

12. Fig. 4A – Why is there a significant reduction in the proliferation with 10 $\mu\text{g/ml}$ ThC compared to 1 $\mu\text{g/ml}$? Is this expected?

Thank you for your comment. We indeed expected this result.

The suppression of thrombopoietin receptors via an excess amount of agonist has been observed in previous studies (Yoshida et al. *Exp. Hematol.* 59 (2018): 30-39., Xie *et al. J. Cell. Mol. Med.* 22.11 (2018): 5367-5377) and reviewed by Hitchcock (*Platelets* 32, 770-778 (2021), Ref 4 of this manuscript). This is likely due to saturation of the ligand-binding domain of a receptor molecule on the cell surface by a ligand, which interferes with ligand binding to another receptor, perturbing receptor dimerization and activation. In addition, ThC has agglutinating activity at high concentrations, which might have a negative effect on cell proliferation. To explain this, we have added the following line to the legend of Figure 4A.

“Note that cell proliferation was suppressed in the presence of 10 $\mu\text{g/mL}$ ThC, presumably due to the suppression of cell proliferation via its agglutinating activity and due to interference with receptor dimerization mediated by the occupation of a ligand-binding site on a receptor molecule.”

13. Fig. 4D – This is interesting and important data – but why switch to HEK cells? It's clear that you have a good Ba/F3-MPL model up and running, why not introduce the mutations in these cells instead? This would allow you to compare more signalling/growth with a direct comparison to your previous data and in a slightly more physiologically relevant cell line. It is likely that the receptor density, diffusion and signalling will be different in the

HEK cells.

We used a reporter assay with HEK cells to investigate N-glycan involvement because this transfection-based assay circumvented the need to establish a cell line and allowed us to examine significantly more diverse mutant forms of MPL in a short time. We agree with your concern regarding the use of different systems to investigate ThC-dependent MPL activation. However, the HEK reporter system has been used to examine MPL ligands, including mutCALR (Shide, *et al. Leukemia* (2017): 1136-1144. Araki *et al. Leukemia* 33.1 (2019): 122-131, Levy *et al. Blood* 135.12 (2020): 948-953), and the consistency between the transient (HEK and other cell lines) and stable (Ba/F3 and other cell lines) expression systems has been recognized. We understand your concern; thus, we describe the reasons for using the HEK system in the Methods section.

“To determine the critical site for ThC activation, mutants in which the N residues were replaced by Q residues were expressed in HEK293T cells, and receptor activation was monitored using the STAT5 reporter assay, which has been recognized as a model system to validate MPL ligands^{21, 36, 37}”

14. Fig. 5A – Can the authors propose a reason why there seems to be a disconnect between pJAK2 and pSTAT5 in the kinetics of ThC activity? pSTAT5 seems to peak at 30mins, which pJAK2 doesn't peak until 360mins. Could this be something to do with the activation of negative regulators? If so, it would be good to identify any changes.

We believe that the different kinetics of pJAK2 and pSTAT5 between TPO and ThC might reflect the availability of phosphatase around JAK2. For TPO, the receptor complex with JAK2 is rapidly internalized, which might induce destruction of the complex and the dephosphorylation of JAK2. For ThC, the receptor complex remained on the cell surface, where the phosphatase might be less abundant and/or suppressed because of the formation of a complex comprising JAK2 with the receptor. In contrast, pSTAT5 is rapidly dephosphorylated, as its regulation is different from the localization of the receptor and JAK2 complex. Furthermore, in the presence of ThC, JAK2 remained on the cell surface along with MPL, and levels of autophosphorylation were slowly increased. Alternatively, the slow and steady activation of JAK2 without the internalization of MPL mediated by ThC might not be associated with the strong phosphorylation of canonical Y1007/1008 residues. Nevertheless, a more detailed analysis is required to elucidate the molecular mechanism underlying JAK2 and downstream activation mediated by lectin ligands.

15. Fig 5B – The internalization data is interesting, but I'm not sure why this method (lysis then membrane protein extraction) was used over flow cytometry. Would this not give more robust and consistent results? Also, the authors cite their previous publication in which this method was used (PMID: 31462733) – but in this paper they show the actual western blot data in addition to the graphs. This would significantly add to the interpretation of this data.

Only a few antibodies are suitable for the flow cytometric analysis of MPL. Furthermore, our pilot experiments suggested potential interference with antibody binding to MPL mediated by the ligand. Therefore, we used a biochemical approach to quantify surface receptor levels. With respect to the western blot data, we have now provided these in supplemental Fig. 14.

16. As the authors propose that ThC uses similar mechanisms to activate MPL and mutCALR, it would be interesting to see if the presence of mutCALR alters the effects of ThC. Presumably there would be competition for binding to MPL?

This is an interesting question. However, to date, we (and perhaps others) have failed to perform a binding assay for mutCALR and MPL with purified proteins. Results of co-immunoprecipitation assay using cell lysate with these proteins co-expressed showed that mutCALR interacts with MPL harboring the immature form of N-glycan, which is predominantly present in the endoplasmic reticulum (ER) (Chachoua, *et al.* *Blood*. 2016;127(10):1325-35, Masubuchi, *et al.* *Leukemia*. 2020;34(2):499-509). These observations suggest that the interaction between mutCALR and MPL requires conditions only present in the ER, which need to be defined to perform the competition assay.

Response to Reviewer #3:

17. Some elaboration as to the process by which the marine sponge was selected to study lectins. Was this part of a broad lectin analysis, or search for thrombopoietins? Were many other organisms part of a screen to identify lectins with thrombopoietic properties

Thank you for your comments. In our previous publication (Ref 12 of the manuscript), we aimed to identify thrombopoietic compounds and screened approximately 900 marine organisms. ThC was found to be an active molecule and one of the most potent extracts. However, the hit rate was low, and only four other active specimens were identified.

18. The lectin isolated by the sponge – does this have any similarity of human F-type lectins? Is there a homologous lectin in humans that may have physiologic consequences?

F-type lectins are ~140 aa lectins with a jelly roll motif forming a trimer. Two trimers are stacked into hexamers to form functional lectins (Vasta *et al. Frontiers in Immunol.* 8 (2017): 1648). This structural motif is very different from that of ThC, which we do not expect to have biological activity like ThC. However, it is interesting to hypothesize that some human lectins act like ThC under normal physiological conditions. It could be a fucose-binding lectin, but lectins with other sugar selectivities might function in mechanisms such as those of CALRmut. The physiological importance of sugar-mediated MPL activation is an intriguing question to pursue.

REVIEWERS' COMMENTS

Reviewer #2 (Remarks to the Author):

My issues have been addresses either through new data or explanations in the text. I am therefore happy to support the publication of this manuscript.

Reviewer #3 (Remarks to the Author):

Hidden pathway for cytokine receptor activation: a marine sponge-derived lectin reveals sugar-mediated thrombopoietin receptor activation

This is a fascinating work that defines a lectin isolated from the marine sponge as a thrombopoietin synergist. The mechanism of action is related to retained thrombopoietin receptor expression and subsequent thrombopoietin binding.

The work is solid with very advanced protein chemical analysis, crystallization, dose response, and signal transduction. Moreover the work identifies an important link of receptor activation in the CALR mutant context by implicating the same N glycosylation site as activating.

This is an exciting study given the breadth of studies that define this unique lectin isolated from the marine sponge and its unusual effect on thrombopoietin signaling. In addition to the insights provided into CALR MPN mutant context, this also affords an opportunity for improved thrombopoietin delivery as a therapeutic.

The resubmitted manuscript adds clarification to issues brought up by reviewers.

Some limitations of this study however still includes the lack of an in vivo system to study physiologic action of thrombocorticin – could the authors discuss whether this is possible to deliver thrombocorticin in perhaps the TPO KO mouse? Why or why not has this been attempted?

Figure 5D – could this be edited to show differences in signal transduction (lightening bolt) with TPO versus THC versus TPO+THC. As is, the cartoon shows equal signal generated, although that is not what the experiments indicate

Response to REVIEWERS' COMMENTS

Here we show our point-by-point response to reviewer#3

Reviewer #3 (Remarks to the Author):

Hidden pathway for cytokine receptor activation: a marine sponge-derived lectin reveals sugar-mediated thrombopoietin receptor activation

This is a fascinating work that defines a lectin isolated from the marine sponge as a thrombopoietin synergist. The mechanism of action is related to retained thrombopoietin receptor expression and subsequent thrombopoietin binding.

The work is solid with very advanced protein chemical analysis, crystallization, dose response, and signal transduction. Moreover the work identifies an important link of receptor activation in the CALR mutant context by implicating the same N glycosylation site as activating.

This is an exciting study given the breadth of studies that define this unique lectin isolated from the marine sponge and its unusual effect on thrombopoietin signaling. In addition to the insights provided into CALR MPN mutant context, this also affords an opportunity for improved thrombopoietin delivery as a therapeutic.

The resubmitted manuscript adds clarification to issues brought up by reviewers.

Some limitations of this study however still includes the lack of an in vivo system to study physiologic action of thrombocortin – could the authors discuss whether this is possible to deliver thrombocortin in perhaps the TPO KO mouse? Why or why not has this been attempted?

Our response

Thank you for your comments.

We have thought about testing efficacy of ThC in vivo. As this reviewer suggested use of TPO knock-out mouse is the most feasible option if we were to conduct the experiment, however, lack of megakaryocytic action of ThC in the iPS cells, likely due to toxicity, suggested that good in vivo action of ThC is not expectable. Moreover, proteinous nature of ThC may result in unfavorable pharmacokinetics and thus require efficient drug delivery system for in vivo application. Blood cell aggregation by the lectin may also

cause adverse effect in vivo also. We therefore concluded in vivo evaluation of this compound is not straightforward and beyond the scope of the paper.

In response to the reviewer's comment, we add a text below in Discussion

Although in vivo efficacy of ThC in mouse models is of great interest, the modification of ThC to reduce its adverse actions is necessary for the experiment.

Figure 5D – could this be edited to show differences in signal transduction (lightening bolt) with TPO versus THC versus TPO+THC. As is, the cartoon shows equal signal generated, although that is not what the experiments indicate

Our response

Thank you for your constructive comment.

We modified figure so that it complies better with our actual data.